# PAC-Bayesian Reinforcement Learning Trains Generalizable Policies

**Abdelkrim Zitouni** [1]  **Mehdi Hennequin** [2]  **Juba Agoun** [1]  **Ryan Horache** [3]  **Nadia Kabachi** [4]  **Omar Rivasplata** [5]

## Abstract

We derive a novel PAC-Bayesian generalization bound for reinforcement learning that explicitly accounts for Markov dependencies in the data, through the chain's mixing time. This contributes to overcoming challenges in obtaining generalization guarantees for reinforcement learning, where the sequential nature of data breaks the independence assumptions underlying classical bounds. The new bound provides non-vacuous certificates for modern off-policy algorithms such as Soft Actor-Critic. We demonstrate the practical utility of the bound through PB-SAC, a novel algorithm that optimizes the bound during training to guide exploration. Experiments across several continuous control tasks show that the proposed approach provides meaningful confidence certificates while maintaining competitive performance.

## 1. Introduction

Deploying reinforcement learning (RL) algorithms in real environments and safety-critical applications requires high confidence that the learned policies will generalize beyond the training data. Although the statistical learning literature has proved rigorous generalization guarantees for supervised learning, and despite progress in generalization guarantees for deep learning models (Pérez-Ortiz et al., 2021) in this setting, some challenges prevent extending to other settings; particularly pertaining to data assumptions.

In RL, an outstanding challenge stems from the fact that algorithms learn from sequential, temporally correlated trajectories that break the standard assumption of independent and identically distributed (*i.i.d.*) data underlying most classical generalization theory. Since RL trajectories exhibit strong temporal dependencies, where future states depend on past actions and the evolving policy, classical analyses that rely on the *i.i.d.* assumption, and their associated sharp concentration bounds, cannot be directly applied to provide meaningful generalization guarantees in this setting.

Facing the aforementioned challenge, the PAC-Bayesian computations framework, which uses PAC-Bayes bounds (McAllester, 1999; Catoni, 2007; Germain et al., 2015; Alquier, 2024) as optimization objectives and for computing high confidence certificates (Pérez-Ortiz et al., 2021) that hold at distribution level, emerges as a promising solution: the analysis maintaining distributions over hypotheses rather than a single model can be extended to temporally dependent data through using appropriate concentration inequalities and new techniques for controlling the exponential moment needed to obtain a new PAC-Bayes bound.

Various approaches have tried to relax the independent data condition, to handle data with dependence structures. Martingale-based methods (Seldin et al., 2011; 2012) extend PAC-Bayesian analysis to sequential settings by constructing martingale sequences from the data (e.g., value function errors, Bellman residuals), then applying martingale concentration inequalities such as Azuma-Hoeffding's (Azuma, 1967) and Freedman's (Freedman, 1975). Although these approaches are mathematically very elegant, RL problems do not naturally yield martingales. Since RL data possess Markov structure by design, concentration inequalities tailored specifically for Markov chains can provide a more natural and better suited approach to this domain.

In this work, we present a novel PAC-Bayesian bound for RL that explicitly accounts for Markov dependencies through the chain's mixing time.[1] Our key technical contribution integrates a bounded-differences condition on the negative empirical return with McDiarmid-type concentration inequality for Markov chains (Paulin, 2018), yielding a bound with explicit constants and improved scaling that avoids the vacuity of previous approaches (cf. Section 2.4). We demonstrate that this new bound is not merely a theoretical curiosity by introducing PB-SAC, an actor-critic algorithm that makes practical use of the bound as a live, optimizable objective.

[1]Université Lumière Lyon 2, Université Lyon 1, ERIC, 69007, Lyon, France [2]Omundu, Lyon, France [3]Université Claude Bernard Lyon 1, LIRIS, UMR CNRS 5205, France [4]Université Lyon 1, Université Lumière Lyon 2, ERIC, 69100, Villeurbanne, France [5]University of Manchester, UK. Correspondence to: Abdelkrim Zitouni <abdelkrim.zitouni@univ-lyon2.fr>.

*Proceedings of the 43$^{rd}$ International Conference on Machine Learning*, Seoul, South Korea. PMLR 306, 2026. Copyright 2026 by the author(s).

---

[1]The smallest number of steps required for the chain's distribution to become nearly indistinguishable from its stationary distribution, regardless of its initial state.

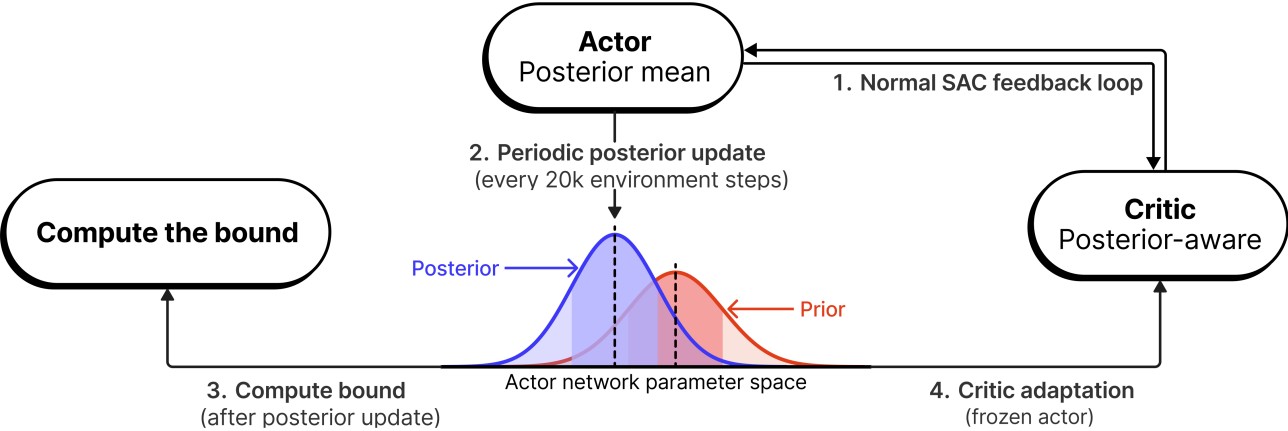

*Figure 1.* High-level illustration of our algorithm (PB-SAC) flow described in Section 4, showing the interaction between the actor, critic, and bound computation components. The periodic posterior update (step 2) guides exploration through PAC-Bayes bounds (step 3) while maintaining standard Soft Actor-Critic (SAC) training (step 1) using an adaptive sampling (step 4).

The algorithm periodically computes the numerical value of the PAC-Bayesian bound and uses it to guide learning through posterior sampling, transforming the generalization guarantee from a passive post-training evaluation tool into an active component of a learning algorithm.

Thus, the work at hand proposes the following contributions. From a theoretical perspective, we develop a PAC-Bayesian bound for RL with explicit mixing-time dependence and improved scaling with realistic trajectory lengths and discount factors relative to prior work (cf. Section 2.4), representing a step forward in statistical learning theory for sequential temporally-dependent data and RL certificates, specifically via PAC-Bayesian computations to obtain tight certificates. Algorithmically, we introduce PB-SAC (Figure 1), the first practical algorithm for Soft Actor-Critic that puts into action a PAC-Bayes bound as an optimizable objective in modern deep RL, including key innovations for stable optimization. Empirically, we demonstrate that bound values remain informative across continuous-control tasks while maintaining competitive performance with state-of-the-art methods.

Our approach establishes the first practical PAC-Bayesian computations framework for optimization and certification in modern RL, bridging the gap between learning theory and algorithmic practice in sequential decision-making.

## 2. Preliminaries

In this section, we briefly recall the definitions of reinforcement learning and statistical learning theory concepts we rely on throughout the paper. The exposition in this section is intentionally concise—the goal is to fix notation and state the learning theory principles that underpin our results.

### 2.1. Reinforcement Learning

Reinforcement Learning (RL) studies how an *agent* learns to make decisions through sequential interaction with an environment. Formally, the environment is modeled as a (possibly unknown) Markov Decision Process (MDP) denoted $\mathcal{M} = (\mathcal{S}, \mathcal{A}, \mathbb{P}, R, \gamma)$; where $\mathcal{S}$ is the state space, $\mathcal{A}$ the action space, $\mathbb{P}(s' \mid s, a)$ the transition kernel, $R(s, a)$ the reward function bounded in $[0, R_{\max}]$, and $\gamma \in (0, 1)$ a discount factor. At each time $t$, the agent observes a state $S_t \in \mathcal{S}$, chooses an action $A_t \in \mathcal{A}$ according to a *policy* $\pi(a|s)$, and then observes a reward $R_{t+1} = R(S_t, A_t)$ and a transition to a new state $S_{t+1} \sim \mathbb{P}(\cdot \mid S_t, A_t)$.

The learning agent's objective is, at least in principle, to maximize the *expected discounted return*

$$V_\pi(s) = \mathbb{E}_{\pi, \mathbb{P}}\big[G_t \mid S_t = s\big], \tag{1}$$

where

$$G_t = \sum_{k=0}^{\infty} \gamma^k R_{t+k+1}. \tag{2}$$

The function $V_\pi(\cdot)$ is the state–value function. It is well known that the optimal value function $V^\star(s) = \sup_\pi V_\pi(s)$ satisfies the Bellman optimality equation

$$V^\star(s) = \max_{a \in \mathcal{A}} \Big\{ R(s, a) + \gamma \, \mathbb{E}_{s' \sim \mathbb{P}}\big[V^\star(s') \mid s, a\big] \Big\}. \tag{3}$$

RL algorithms either learn directly a policy (policy–gradient and actor–critic methods, see e.g. Haarnoja et al. (2018); Konda & Tsitsiklis (1999); Sutton et al. (1999)), or learn an action–value function $Q_\pi(s, a)$ in value–based methods such as Q-learning and its deep variants (Mnih et al., 2013).

Model–free approaches do not use an explicit model of $\mathbb{P}$, while model–based methods leverage or learn a transition model to plan (Sutton & Barto, 2018).

## 2.2. Probably Approximately Correct (PAC) Bounds

In a supervised learning task, the instance space is taken to be the product $\mathcal{Z} = \mathcal{X} \times \mathcal{Y}$, where $\mathcal{X} \subseteq \mathbb{R}^d$ is the feature space and $\mathcal{Y}$ the label space; commonly $\mathcal{Y} \subseteq \mathbb{R}$ for regression, and $\mathcal{Y} \subseteq \mathbb{N}$ for classification problems. In this setting, a learning algorithm is a function that takes in a training sample $S = \{(\boldsymbol{x}_i, y_i)\}_{i=1}^m$ and returns a prediction function $f_\theta : \mathcal{X} \to \mathcal{Y}$, also referred to as a hypothesis, parametrized by $\theta \in \Theta$, where $\Theta \subset \mathbb{R}^p$ denotes the set of all admissible parameter vectors. An unknown data distribution $\mathcal{D}$ over $\mathcal{Z}$ is postulated, with $\mathcal{D}_\mathcal{X}$ denoting the marginal distribution on $\mathcal{X}$, and training samples $S = \{(\boldsymbol{x}_i, y_i)\}_{i=1}^m$ are such that each pair $(\boldsymbol{x}_i, y_i) \in \mathcal{Z}$ is an independent and identically distributed (i.i.d.) random draw from $\mathcal{D}$, that is, $S \sim \mathcal{D}^{\otimes m} := \mathcal{D} \otimes \cdots \otimes \mathcal{D}$ ($m$ copies). The "quality" of a hypothesis $f_\theta$ is typically assessed through a loss function $\ell : \mathcal{Y} \times \mathcal{Y} \to \mathbb{R}_+$, which quantifies the discrepancy between predicted and true outputs (labels). Two important functionals associated to a hypothesis are its *true risk* on the distribution $\mathcal{D}$, and its *empirical risk* on the sample $S$, which are respectively given by

$$\mathcal{L}(\theta) = \mathbb{E}_{(x,y)\sim\mathcal{D}}\big[\ell\big(f_\theta(\boldsymbol{x}), y\big)\big], \qquad (4)$$

$$\hat{\mathcal{L}}_S(\theta) = \frac{1}{m}\sum_{i=1}^m \ell\big(f_\theta(\boldsymbol{x}_i), y_i\big). \qquad (5)$$

In supervised machine learning, the goal is to learn a hypothesis $f_\theta$ that accurately predicts labels $y \in \mathcal{Y}$ for given inputs $\boldsymbol{x} \in \mathcal{X}$. Since training is based on a finite dataset, a central question is: how can we ensure that the learned function $f_\theta$ will perform well on unseen data?

Probably Approximately Correct (PAC) learning answers this question via probability inequalities saying that, under suitable restrictions, for a chosen level $\delta$ within $(0, 1)$ we have

$$\Pr_{S\sim\mathcal{D}^m}\big\{\mathcal{L}(\theta) \leq \hat{\mathcal{L}}_S(\theta) + \epsilon\big\} \geq 1 - \delta.$$

Choosing a small $\delta$ then translates into a high-confidence bound for $\mathcal{L}(\theta)$. Concrete PAC bounds specify how large $m$ must be (or how large the gap $\epsilon$ can be) in terms of properties of the hypothesis class, e.g. VC-dimension, Rademacher complexity, stability, compression, etc.

All of the classical PAC bounds treated $f_\theta$ as a deterministic output of the learning algorithm.

## 2.3. PAC-Bayesian Bounds

The PAC-Bayesian literature (see e.g. Alquier, 2024, for a comprehensive coverage) extends PAC learning bounds to

analyze distributions over hypothesis, rather than individual hypotheses. PAC-Bayes bounds have been useful in studying generalization for stochastic learning algorithms and prediction rules based on randomizing or averaging. Let $\Theta$ denote the set of parameters defining a family of prediction functions $\{f_\theta : \mathcal{X} \to \mathcal{Y}\}_{\theta\in\Theta}$. A distribution $\mu \in \mathcal{P}(\Theta)$ is specified over $\Theta$, called a *prior* distribution to indicate being completely or largely independent of data (see below). Upon receiving data $S \sim \mathcal{D}^{\otimes m}$, the learning algorithm then selects a *posterior* distribution $\rho \in \mathcal{P}(\Theta)$. PAC-Bayesian theory provides high-confidence bounds on the population risk $\mathbb{E}_{\theta\sim\rho}[\mathcal{L}(\theta)]$ in terms of the empirical risk $\mathbb{E}_{\theta\sim\rho}[\hat{\mathcal{L}}_S(\theta)]$ and an additional term that effectively constrains the complexity of the posterior distribution $\rho$ via an information quantity measuring the discrepancy between $\rho$ and $\mu$, typically the Kullback-Leibler divergence $\mathrm{KL}(\rho\|\mu)$, but there are other choices. Formally, for any $\kappa > 0$ and chosen level $\delta \in (0, 1)$, the following inequality holds with probability of at least $1 - \delta$ over the random draw of the training sample $S$, simultaneously for all distributions $\rho$:

$$\mathbb{E}_{\theta\sim\rho}[\mathcal{L}(\theta)] \leq \mathbb{E}_{\theta\sim\rho}[\hat{\mathcal{L}}_S(\theta)]$$
$$+ \frac{1}{\kappa}\Big(\mathrm{KL}(\rho\|\mu) + \ln\frac{1}{\delta} + \Psi_{\ell,\mu}(\kappa, m)\Big) \qquad (6)$$

with

$$\Psi_{\ell,\mu}(\kappa, m) = \ln \mathbb{E}_{\theta\sim\mu} \mathbb{E}_{S\sim\mathcal{D}^{\otimes m}}\Big[\exp\big(\kappa\big(\mathcal{L}(\theta) - \hat{\mathcal{L}}_S(\theta)\big)\big)\Big].$$

Compared with classical PAC guarantees, PAC-Bayes offers two advantages that are critical for reinforcement learning: (1) *Data-dependent priors (Parrado-Hernández et al., 2012)* —when $\mu$ can itself depend on previous data, e.g. earlier tasks or behavioural trajectories (Zhang et al., 2025), the bound adapts to the knowledge already acquired, tightening $\mathrm{KL}(\rho\|\mu)$; and (2) *Fine-grained control via $\Psi$* —by tailoring the concentration inequality used to upper-bound $\Psi$ one can incorporate dependence structures such as martingales (Seldin et al., 2012; 2011), $\beta$-mixing (Ralaivola et al., 2009; Abélès et al., 2025) sequences or Markov chains (Fard et al., 2011; Tasdighi et al., 2025). The latter is exactly the scenario in which RL trajectories are collected.

## 2.4. Previous PAC-Bayesian Bounds in RL

Early efforts to apply PAC-Bayesian theory to RL, notably the works of Fard & Pineau (2010) and Fard et al. (2011), established the framework's viability for model selection in batch settings. However, their bounds suffered from poor scaling with the discount factor, rendering them numerically vacuous for problems with long effective horizons typical in modern RL (Tasdighi et al., 2025). Contemporary approaches have repurposed PAC-Bayes bounds for other algorithmic goals. Recent works have used them to derive training objectives for deep exploration (Tasdighi

et al., 2025) or as regularizers for lifelong learning (Zhang et al., 2025). While these works demonstrate the versatility of PAC-Bayesian computations for algorithm design, they sidestep the original goal of providing tight, computable certificates for modern deep RL agents. This reveals a fundamental gap: although PAC-Bayesian theory holds promise for certified RL, existing approaches either yield vacuous bounds or repurpose them for different objectives. To the best of our knowledge, no practical algorithm had leveraged non-vacuous PAC-Bayesian bounds as live performance certificates within modern RL frameworks to date.

## 3. A new PAC-Bayesian Bound for RL

We now present our main theoretical contribution: a new PAC-Bayes generalization bound for RL that explicitly accounts for temporal dependencies in data trajectories through the mixing time of the underlying Markov chain.

### 3.1. Problem Setup

As outlined earlier, our objective is to establish a *high-probability* PAC-Bayes **value-error** bound for a policy operating in a Markov decision process (MDP) when the training data are *dependent* trajectories—possibly gathered under an off-policy algorithm. In this section, we begin by fixing notation and then present the main results; all proofs are deferred to Appendix B.

Let $\mathcal{M} = (\mathcal{S}, \mathcal{A}, \mathbb{P}, R, \gamma)$ be a discounted MDP, where $\mathcal{S}$ and $\mathcal{A}$ are the state and action spaces, $\mathbb{P}$ is the transition kernel, $R$ is the reward function such that $R_t \in [0, R_{\max}]$, and $\gamma \in (0, 1)$ is the discount factor. A policy $\pi_\theta$ induces a (not necessarily *time-homogeneous*) Markov chain $\xi = (S_1, A_1, R_1, S_2, \dots)$ terminating at state $S_H$ for trajectories of finite horizon $H$ considered here. Our analysis naturally extends to the infinite-horizon case. The distribution of this chain is determined by the initial state distribution $\nu$, transition probabilities $\mathbb{P}$, stochastic policy $\pi_\theta$, and random rewards $R$ as described next.

We assume access to a dataset $\mathfrak{D} = \{\xi^{(1)}, \dots, \xi^{(T)}\}$ of $T$ trajectories (i.e., $N = HT$ transitions in total), collected using a behavior policy $\pi_b$. An evaluation policy $\pi_\theta$ is parameterized by $\theta \in \Theta$, the parameters $\theta$ are drawn from a distribution $\rho \in \mathcal{P}(\Theta)$, where $\Pi = \{\pi_\theta : \theta \in \Theta\}$ denotes the policy class. Henceforth, we write $\xi \sim \mathcal{M}$ (**resp.** $\mathfrak{D} \sim \mathcal{M}^{(T)}$) to denote sampling a trajectory (**resp.** a set $\mathfrak{D}$ of $T$ trajectories) under the environment dynamics $\mathbb{P}$, initial state distribution $\nu$, policy $\pi_b$, and reward function $R$, in order to avoid notational overload. Throughout, *dependence* refers strictly to *intra-trajectory* dependence: the transitions within a single episode $(S_1, A_1, R_1, S_2, \dots, S_H)$ are sequentially correlated through the Markov property, since each $S_{t+1}$ is determined by $(S_t, A_t)$.

We define the discounted return of a trajectory $\xi$ and the value of policy $\pi_\theta$ respectively as:

$$G(\xi) = \sum_{k=0}^{H-1} \gamma^k R_{k+1} \quad \text{and} \quad V_{\pi_\theta} = \mathbb{E}_{\xi \sim \mathcal{M}}[G(\xi)] \quad (7)$$

We now define the expected (true) loss and its empirical counterpart using importance sampling to account for the off-policy data distribution:

$$\mathcal{L}(\theta) = -\mathop{\mathbb{E}}_{\xi \sim \mathcal{M}} \left[ \frac{\pi_\theta(\xi)}{\pi_b(\xi)} G(\xi) \right] \quad (8)$$

$$\hat{\mathcal{L}}_{\mathfrak{D}}(\theta) = -\frac{1}{T} \sum_{j=1}^{T} w_j(\theta) G(\xi^{(j)}), \quad (9)$$

where $w_j(\theta) = \frac{\pi_\theta(\xi^{(j)})}{\pi_b(\xi^{(j)})}$ is the likelihood ratio of trajectory $\xi^{(j)} \sim \pi_b$. Notice that the estimator $\hat{\mathcal{L}}_{\mathfrak{D}}(\theta)$ is unbiased:

$$\mathcal{L}(\theta) = \mathop{\mathbb{E}}_{\mathfrak{D} \sim \mathcal{M}^{(T)}}[\hat{\mathcal{L}}_{\mathfrak{D}}(\theta)].$$

To ensure computational stability, we assume the weighted rewards are bounded such that the weighted return remains within the range consistent with $R_{\max}$ (e.g., via clipping large importance weights, a standard practice in off-policy evaluation).

*Remark* 3.1 (Weight clipping preserves the certificate). Note that clipping the importance weights as $w_j^c = \min(w_j, M)$ introduces bias, but this bias is strictly *pessimistic* and does not invalidate the bound. Since $G(\xi) \geq 0$ and $w_j^c \leq w_j$, the clipped true loss satisfies $\mathcal{L}(\theta) \leq \mathcal{L}_c(\theta)$, where $\mathcal{L}_c(\theta) = -\mathbb{E}[w^c G(\xi)]$. Applying Theorem 3.3 to $\mathcal{L}_c$ and chaining gives $\mathcal{L}(\theta) \leq \hat{\mathcal{L}}_{\mathfrak{D}}^c(\theta) + \text{complexity term}$, so the certificate remains formally valid, albeit more conservative. A complete derivation is provided in Appendix B.2.

Following PAC-Bayesian folklore, we endow the parameter space $\Theta$ with a distribution $\mu \in \mathcal{P}(\Theta)$ playing the role of a prior selected independently of data (or partial dependence on it, cf. Parrado-Hernández et al. (2012); Pérez-Ortiz et al. (2021)), and a posterior $\rho \in \mathcal{P}(\Theta)$ chosen after observing $\mathfrak{D}$. This formalism enables reasoning about randomized policies drawn from $\rho$ with guarantees based on their divergence from $\mu$. Crucially for our analysis, changing one transition in the data results in a quantifiable bounded effect on the empirical loss defined in (8):

**Lemma 3.2** (Bounded differences). *Let $\theta \in \Theta$ be fixed policy parameters, and let $\mathfrak{D}$ and $\bar{\mathfrak{D}}$ be sets of trajectories. Then, there exists $c \in \mathbb{R}_+^{H \times T}$ such that*

$$\left| \hat{\mathcal{L}}_{\mathfrak{D}}(\theta) - \hat{\mathcal{L}}_{\bar{\mathfrak{D}}}(\theta) \right| \leq \sum_{h'=1}^{H} \sum_{j'=1}^{T} c_{(h',j')} \mathbb{I}\left[ \xi_{h'}^{(j')} \neq \bar{\xi}_{h'}^{(j')} \right] \quad (10)$$

Intuitively, $c_{(h,j)}$ quantifies the *transition-level influence* of altering the $(h, j)$-th state–action–reward tuple on the average return. A complete derivation—including a justification of why this bound covers propagation of the perturbed transition to future steps—is given in Appendix B.2. The result yields the explicit vector $c \in \mathbb{R}_+^{H \times T}$ to be used, namely

$$c_{(h,j)} = \frac{\gamma^{h-1} R_{\max}}{T}, \qquad (11)$$

and its norm (Appendix B.3) used in Theorem 3.3 below:

$$\|c\|^2 = \frac{R_{\max}^2}{T(1-\gamma^2)}\left(1 - \gamma^{2H}\right). \qquad (12)$$

### 3.2. Main Result

The above bounded-differences property (Lemma 3.2) is precisely what allows us to apply concentration inequalities to dependent data. Importantly, the proof strategy is leveraging Paulin (2018)'s extension of McDiarmid's inequality to Markov chains, which provides concentration for functions satisfying bounded differences on Markovian sequences. The key insight is that while the transitions are temporally dependent, the bounded-differences condition with explicit constants $\|c\|^2$ enables us to control how perturbations propagate through the dependency structure.

Applying Paulin (2018)'s concentration result yields a tail bound on the deviation $\mathcal{L}(\theta) - \hat{\mathcal{L}}_{\mathfrak{D}}(\theta)$ (see inequality (21)) that depends explicitly on the mixing time $\tau_{\min}$ of the policy-induced Markov chain. $\tau_{\min}$ is the smallest number of steps after which the distribution of the chain's state is, in a statistical sense, nearly indistinguishable from its long-run or stationary distribution in total variation distance, no matter where the chain started. In other words, it measures how quickly the chain "forgets" its initial state and becomes well mixed. Combining the tail bound from inequality (21) with the standard PAC-Bayesian change-of-measure technique gives our main result:

**Theorem 3.3.** *Let the weighted reward function be bounded in $[0, R_{max}]$ and let $\mathcal{M}$ be a (not necessarily time-homogeneous) Markov Decision Process (MDP) induced by any policy $\pi_b$ such that it satisfies $\tau_{\min} < +\infty$. For any prior $\mu$ over $\Theta$, and any $\delta \in (0, 1)$, with probability at least $1 - \delta$ over the sample $\mathfrak{D}$ of $T$ trajectories with time horizon $H$, simultaneously for all posterior distributions $\rho$ over $\Theta$, we have:*

$$\mathbb{E}_{\theta \sim \rho}[\mathcal{L}(\theta)] \leq \mathbb{E}_{\theta \sim \rho}[\hat{\mathcal{L}}_{\mathfrak{D}}(\theta)]$$

$$+ \sqrt{\frac{R_{\max}^2(1-\gamma^{2H})}{T(1-\gamma^2)}\tau_{\min}\left(\mathrm{KL}(\rho\|\mu) + \ln\frac{\sqrt{2}}{\delta}\right)}. \quad (13)$$

In particular, this inequality can be applied to posteriors $\rho$ chosen after interacting with the environment.

The bound in Theorem 3.3 can be straightforwardly converted to a PAC-Bayes lower bound on the true expected value function $\mathbb{E}_{\theta \sim \rho}[V_{\pi_\theta}]$ (see Appendix B, inequality (33)), using the fact that $\mathcal{L}(\theta) = -V_{\pi_\theta}$ ((8), (9)) and a simple rearrangement of terms. The true expected value is lower-bounded by an empirical estimate minus an uncertainty term that accounts for limited data ($1/T$), temporal correlations ($\tau_{\min}$), and posterior complexity ($\mathrm{KL}(\rho\|\mu)$). This interpretation suggests a natural approach to policy optimization: select the posterior $\rho$ that maximizes this lower bound (equivalently, minimizes the upper bound in inequality (13)). Such a strategy would automatically balance exploitation (maximizing the empirical value) and theoretically-justified exploration (accounting for uncertainty).

### 3.3. Key Improvements and Discussion

**Improved Horizon Dependence.** Previous PAC-Bayes bounds for RL suffered from prohibitive scaling with the effective horizon $1/(1-\gamma)$. Fard et al. (2011) bounded the value error via the Bellman error, a conversion (their Eq. (7)) that degrades sample complexity to $\mathcal{O}((1-\gamma)^{-4})$. Tasdighi et al. (2025) face comparable constraints. For $\gamma = 0.99$, such scaling renders these bounds practically vacuous. In contrast, our transition-level analysis bounds the value error directly via the concentration of discounted returns, achieving a scaling of $\mathcal{O}((1-\gamma)^{-1})$, significantly tighter. This improvement arises because our sensitivity term scales with $\sum^H \gamma^{2h} \approx (1-\gamma)^{-1}$ (Appendix B.2), avoiding the $(1-\gamma)^{-2}$ penalty inherent to the two-step derivation from the Bellman error. Specifically, we require only $T \gtrsim \frac{R_{\max}^2 \tau_{\min}}{1-\gamma^2}$ trajectories.

**Explicit Mixing Time Dependence.** Unlike bounds for general mixing processes that depend on abstract coefficients, our result features explicit dependence on $\tau_{\min}$, the mixing time of the policy-induced Markov chain. This quantity is well studied and has a clear interpretation in terms of environment dynamics. While extending concentration inequalities from independent to mixing settings is sometimes viewed as straightforward, the reality involves several subtle challenges. Although the distribution of $X_t$ becomes close to the stationary distribution $\pi$ after $\tau_{\min}$ steps, this does not guarantee that $X_t$ is approximately independent of $X_0$, let alone that consecutive states $X_t$ and $X_{t+1}$ are independent. The dependence between observations decays exponentially with the mixing time, but achieving approximate independence typically requires waiting several multiples of $\tau_{\min}$, not just $\tau_{\min}$ itself.

Moreover, applying McDiarmid-type inequalities to RL requires establishing that the negative empirical return satisfies a bounded-differences condition with explicit, tractable constants (Lemma 3.2). This necessitates careful analysis of how perturbations at individual transitions propagate

through the sequential structure to affect future states and rewards. Our transition-level analysis directly addresses this challenge by quantifying the error propagation through the Markov dependency structure. The proof is provided in Appendix B.4.

**Practical Tractability and Robustness.** The bound requires estimating $\tau_{\min}$, which presents both computational and robustness considerations. We estimate mixing time using autocorrelation decay of the reward signal, as this can be computed from streaming trajectories without storing full visitation counts. An alternative approach utilizing the pseudo-spectral gap (Karagulyan & Alquier, 2025) could offer tighter bounds when the gap can be estimated. However, constructing fully empirical estimators for these gaps currently requires finite state spaces or specific parametric assumptions (e.g., AR(1)), rendering them intractable for general deep RL in continuous environments.

The estimation approach used here is robust to errors in a specific direction: if we overestimate $\tau_{\min}$, our bound remains valid but becomes looser, which is harmless in practice. However, underestimation can be problematic as it leads to overconfidence. To mitigate this, we compute autocorrelation from additional sources (e.g., state features) to cross-validate mixing time estimates. In practice, we err on the side of caution by using a conservative initial estimate and taking the maximum with the latest autocorrelation estimation, trading some tightness for reliability. Empirical analysis can be found in Section 5.4.

# 4. PB-SAC: A Practical Algorithm for PAC-Bayesian Reinforcement Learning

Translating the theoretical PAC-Bayes bound into a practical learning algorithm requires addressing challenges of maintaining posterior distributions over policy parameters in deep RL. The novel algorithm proposed here, **PAC-Bayes Soft Actor-Critic (PB-SAC)**, builds upon SAC while integrating PAC-Bayesian bounds and posterior-guided exploration (pseudo-code and illustrative figure in Appendix F).

**Motivation and notion of safety.** The primary motivation for PB-SAC is *deployment safety*: providing a formal guarantee that a trained policy will perform well on unseen trajectories, not merely on those encountered during training. This notion of safety is distinct from "Safe RL" in the obstacle-avoidance sense; it concerns the *reliability of empirical performance estimates*. Standard SAC can achieve high training scores while silently overfitting with no mechanism to detect the gap before deployment. PB-SAC addresses this by computing a rigorous PAC-Bayesian lower bound on the true expected return throughout training (solid lines in Figure 2b), which acts as a deployment certificate. A *small gap* between the empirical return and this lower bound is evidence that the empirical estimate reliably predicts future performance on unseen data; a *large gap* is a quantifiable warning against deployment. Actively optimizing this lower bound—rather than treating it as a passive post-hoc tool—allows PB-SAC to transform the generalization guarantee into a live component of the learning process.

PB-SAC's central insight is that policy parameters always represent the posterior mean, updated through standard SAC gradients during regular training, ensuring most learning follows proven SAC dynamics while periodic PAC-Bayesian updates refine the posterior and guide exploration. Following Zhang et al. (2025), we maintain a diagonal Gaussian posterior $\rho(\theta) = \mathcal{N}(\upsilon, \text{diag}(\sigma^2))$ over flattened policy parameters with learnable mean $\upsilon$ and standard deviation $\sigma$. The prior $\mu$ undergoes periodic moving average updates toward the current posterior with linear decay, preventing KL divergence explosion while preserving bound validity and maintaining exploration capability as the prior stabilizes during training.

## 4.1. Posterior-Guided Exploration

During exploration, PB-SAC leverages the posterior distribution to implement uncertainty-driven exploration. Rather than standard $\epsilon$-greedy exploration, the algorithm samples policies from the posterior and selects actions that maximize Q-values under posterior uncertainty:

$$\theta_{explore} \leftarrow \arg\max_{\theta_i \in \mathfrak{T}} Q(s, \pi_\theta(s))$$

such that $\mathfrak{T}$ is a set of policy parameter vectors drawn from $\rho$. Crucially, $\mathfrak{T}$ is a *finite* sample: at each exploration step we draw $|\mathfrak{T}|$ candidate parameter vectors from $\rho$ and evaluate $Q(s, \pi_{\theta_i}(s))$ for each via a single forward pass through the frozen critic, then take the argmax. The optimisation is therefore $\mathcal{O}(|\mathfrak{T}|)$ critic evaluations—no continuous search over $\Theta$ is required, making PGE fully tractable. The posterior standard deviation $\sigma$ governs the diversity of candidates: a wider posterior encourages exploration of more distant parameter regions, while a narrow posterior concentrates candidates near the current mean, naturally interpolating between exploration and exploitation as training progresses. This posterior-guided exploration naturally balances exploitation of the mean policy (the current actor) when it yields the highest value, with exploration of alternative policies in regions of high posterior uncertainty where potentially superior policies may exist. This approach provides theoretical grounding for the exploration strategy through PAC-Bayesian computations, ensuring that exploration is guided by uncertainty quantification rather than arbitrary randomness.

## 4.2. Alternating Optimization via PAC-Bayes-$\lambda$

The core challenge in optimizing our PAC-Bayesian bound (Theorem 3.3) lies in its structure: while the KL divergence term $KL(\rho\|\mu)$ is convex in its first argument (the posterior parameters), the square root composition in the bound is not guaranteed to be convex, potentially leading to optimization difficulties. To address this challenge, we use variational approximation. By applying the identity $\sqrt{x} = \inf_{\lambda>0}(\frac{x}{2\lambda} + \frac{\lambda}{2})$, we recover the convex objective $\mathcal{J}(\rho, \lambda)$ in (14). This objective has a structure remarkably similar to the PAC-Bayes-$\lambda$ bound of Thiemann et al. (2017), derived from the classical PAC-Bayes-kl bound for majority vote learning (see, e.g., Germain et al. (2015), Corollary 21; Seeger (2003); Langford (2005)). Both approaches share the key insight of introducing an auxiliary trade-off parameter $\lambda$ that transforms non-convex objectives into quasi-convex relaxations amenable to stable alternating optimization.

Our implementation optimizes the PAC-Bayes-$\lambda$ objective shown next to obtain the posterior $\rho$, and then substitutes this optimized posterior into inequality (13).

$$\underbrace{\mathbb{E}_{\theta\sim\rho}[\hat{\mathcal{L}}_{\mathfrak{D}}(\theta)] + \frac{\|c\|^2\tau_{\min}\left(KL(\rho\|\mu) + \ln\frac{\sqrt{2}}{\delta}\right)}{2\lambda} + \frac{\lambda}{2}}_{\mathcal{J}(\rho, \lambda)} \quad (14)$$

This decomposition enables stable alternating optimization: we optimize posterior parameters for fixed $\lambda$, then compute the optimal closed form $\lambda^*$ for fixed posterior parameters (see equation (37)), ensuring convergence and maintaining the theoretical guarantee of our PAC-Bayesian bound. As shown in Appendix B.5.4, optimizing this objective yields a tighter certificate.

## 4.3. Policy-level REINFORCE trick

Even with the above decomposition, we cannot straightforwardly compute the gradient

$$\nabla_{(v,\sigma)}\mathbb{E}_{\theta\sim\rho}\left[\hat{\mathcal{L}}_{\mathfrak{D}}(\theta)\right]$$

because sampling cannot occur inside the gradient operation. To address this challenge, we employ a two-stage approach. First, we collect fresh rollouts during the PAC-Bayes update cycle using the mean policy, then sample policies from $\rho$ and evaluate their discounted returns on these rollouts using importance sampling. Importantly, these rollouts are evaluated as *intact trajectories*—not as randomly shuffled transitions from the replay buffer—so that the intra-trajectory dependence structure required by Theorem 3.3 is preserved (see Section 3.1). This ensures that bound computation accounts for distributional shift between the data-generating policy and the current posterior distribution. With estimated returns for each sampled policy, we apply the log-likelihood

trick (REINFORCE, Williams (1992)) at the policy level rather than the traditional action level. We prove this extension in Appendix E. This technique allows us to exchange gradient and expectation operations:

$$\mathbb{E}_{\theta\sim\rho}\left[\nabla_{(v,\sigma)}\log\mathbb{P}_{v,\sigma}(\theta)\hat{\mathcal{L}}_{\mathfrak{D}}(\theta)\right],$$

yielding a tractable sampling-based gradient estimator.

## 4.4. Adaptive Sampling

Our most critical innovation addresses the actor-critic misalignment problem that arises after PAC-Bayesian updates. When the posterior mean shifts significantly, critics become misaligned with the new policy distribution, creating instability in training. Without this mechanism, ablation studies (Figure 8a) reveal a characteristic sawtooth pattern: performance drops sharply immediately after each PAC-Bayesian update, then gradually recovers until the next update cycle.

PB-SAC resolves this misalignment issue through adaptive sampling: immediately following PAC-Bayesian updates, we freeze the actor and employ high-rate posterior sampling (256 samples) to expose critics to the full posterior distribution for recalibration, then resume efficient learning with minimal sampling (1 sample, the posterior mean), achieving computational efficiency without sacrificing stability.

# 5. Experiments

We now turn to empirical validation across representative continuous control tasks to demonstrate that PAC-Bayesian computations for RL deliver on this promise while maintaining competitive learning performance.

## 5.1. Experimental Design

We evaluate PB-SAC on four MuJoCo continuous control environments (Towers et al., 2024; Tassa et al., 2018) spanning different complexity levels: HalfCheetah, Ant, Hopper, and Walker2d. Our evaluation tracks two critical metrics: PAC-Bayesian certificate evolution (Figure 2, right) and learning performance relative to baselines (Figure 2, left).

We run PB-SAC with PAC-Bayesian updates every 20,000 environment steps, employing our adaptive sampling (256 posterior samples during critic adaptation, posterior mean alone during regular training). We compare against vanilla SAC (Haarnoja et al., 2018) using identical network architectures and include PBAC (Tasdighi et al., 2025), a PAC-Bayes deep exploration method, despite its primary strengths manifesting in sparse reward settings. We justify this comparison by evaluating our algorithm in the same sparse-reward setting, with results reported in Appendix G. Additional ablation studies examining the impact of our algorithmic components (adaptive sampling, posterior-guided

exploration, and mixing time estimation) are also provided in Appendix G. Table 1 summarizes all hyperparameters.

## 5.2. The Tightening of Performance Certificates

Figure 2b reveals the most compelling aspect of our results: the PAC-Bayesian bounds consistently tighten throughout training, tracking the improvement in learned policies. In HalfCheetah, the bounds become informative within $100k$ steps and continue tightening as performance improves, demonstrating that our certificates provide genuine confidence estimates for high-performing policies rather than vacuous guarantees. Particularly noteworthy is the bounds' behavior during performance fluctuations: they appropriately widen during periods of high variance whilst tightening when performance stabilizes (e.g., HalfCheetah between $100k$ and $200k$ steps). This stability stems from decaying moving average prior updates, which prevent KL explosion in early stages; without this mechanism, KL explosion hinders learning by pulling the posterior back towards the prior rather than allowing improvement.

Hopper, Walker2d (Figure 5), and Ant exhibit similar bound evolution, with certificates tracking performance and reflecting meaningful task completion. Notably, Ant—the most challenging test case due to its 3D dynamics and large state space—demonstrates that our bounds remain meaningful even for complex, high-dimensional continuous control tasks. The bounds capture the qualitative behavior predicted by our theory.

## 5.3. Empirical Validation Across Environments

The learning curves in Figure 2a address a fundamental concern about PAC-Bayesian RL: whether theoretical guarantees necessitate performance sacrifices. Our results demonstrate that PB-SAC maintains competitive performance with SAC across all tested environments while providing formal certificates. This validates our architectural choices, particularly the policy-posterior synchronization that ensures most learning follows proven SAC dynamics, with periodic PAC-Bayesian updates serving as refinements rather than disruptions, guided by the adaptive sampling that prevents critic destabilization (see Figure 8a).

PBAC, designed specifically for deep exploration through multi-objective optimization (diversity, coherence, and propagation), underperforms both methods on dense-reward tasks. To ensure a fair comparison—since PBAC's primary strengths manifest in sparse-reward settings—we evaluated PB-SAC on the sparse-reward task Ant (very delayed) from PBAC's original work (Tasdighi et al., 2025). Results (Appendix G.1) demonstrate that PB-SAC ultimately surpasses PBAC while outperforming vanilla SAC, confirming that our posterior-guided exploration provides meaningful benefits in sparse-reward settings without requiring PBAC's high

computational complexity (ensemble of critic networks). These results establish that our framework provides non-vacuous certificates and competitive performance across both dense and sparse reward tasks.

## 5.4. Mixing Time Analysis

A critical component of the new framework is mixing time estimation and its effect on bound evaluation.

Underestimation of $\tau_{\min}$ can lead to overconfident bounds. To assess robustness, we conducted experiments with fixed mixing time estimates ranging from 1 (fastest mixing, potential underestimation) to 1000 (conservative overestimation for MuJoCo tasks) without dynamic updates during training. Figure 7 illustrates these results on HalfCheetah. As expected, smaller mixing time values produce tighter bounds,[2] while larger values yield more conservative certificates. However, a notable finding is that overestimation proves less problematic than anticipated: while the bound remains theoretically valid but looser, the algorithm effectively adapts to this conservatism, maintaining meaningful certificates even with substantial overestimation. This robustness suggests that erring on the side of caution with conservative mixing time estimates is a viable practical strategy.

In practice, we mitigate underestimation risk by cross-checking autocorrelation estimates from multiple signals (rewards and state features) and enforcing monotonicity by taking the running maximum across updates. These results demonstrate that our framework provides reliable certificates across a wide range of mixing time specifications.

# 6. Conclusion

This work addresses a fundamental challenge at the intersection of learning theory and deep reinforcement learning: deriving non-vacuous generalization guarantees for sequential decision-making under temporally correlated data, and turning those guarantees into a live training objective.

Three challenges had to be overcome. First, RL trajectories are Markov-dependent, breaking classical i.i.d. assumptions; a bounded-differences analysis of the empirical return (Lemma 3.2) integrated with Paulin's concentration inequality for Markov chains (Paulin, 2018) addresses this, with the bound carrying explicit dependence on the mixing time $\tau_{\min}$. Second, prior bounds scaled as $\mathcal{O}((1-\gamma)^{-4})$, rendering them vacuous in practice; a direct bound on the value error through discounted returns, rather than through the Bellman error, brings this to $\mathcal{O}((1-\gamma)^{-1})$. Third, the PAC-Bayes objective is non-convex, and periodic posterior

---

[2]Using a mixing time equal to 1 is equivalent to an i.i.d. bound, more specifically to one that can be derived from the original McDiarmid's inequality (McDiarmid, 1989) (not the one we used). This demonstrates how tight such bounds can be.

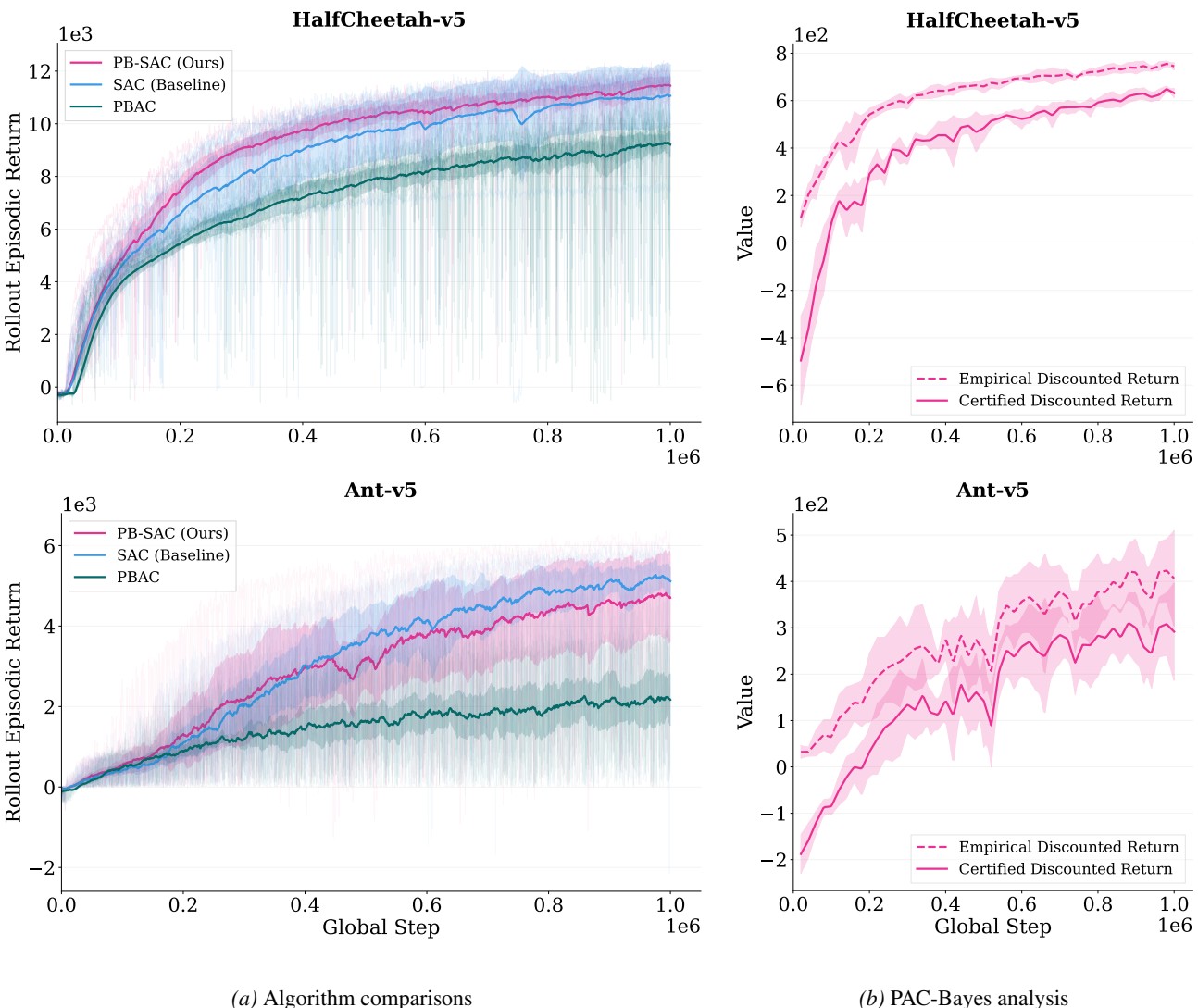

*(a)* Algorithm comparisons

*(b)* PAC-Bayes analysis

*Figure 2.* **(a)** Performance comparison between our ■ PB-SAC, its baseline ■ SAC, and ■ PBAC from Tasdighi et al. (2025); **(b)** PAC-Bayes analysis of PB-SAC across environments. The empirical discounted return (dashed line) corresponds to $\mathbb{E}_{\theta \sim \rho}[-\hat{\mathcal{L}}_{\mathfrak{D}}(\theta)]$, and the certified discounted return (solid line) corresponds to the lower bound on $\mathbb{E}_{\theta \sim \rho}[-\mathcal{L}(\theta)]$ provided by Theorem 3.3 (after rearranging the terms).

updates destabilize the critic. A PAC-Bayes-$\lambda$ variational relaxation converts the objective into a convex alternating-optimization scheme (Section 4); our adaptive sampling mechanism then exposes the critic to the updated posterior at high sample rate after each update, and the sawtooth instability documented in Figure 8a disappears.

PB-SAC, the resulting algorithm, maintains competitive performance with its baseline SAC across dense-reward continuous-control tasks and provides formal lower bounds on the expected return that tighten throughout training. In sparse-reward settings, posterior-guided exploration led to PB-SAC outperforming both SAC and PBAC (Tasdighi et al., 2025), showing that the certification benefit comes with a meaningful exploration advantage.

Several limitations warrant consideration. Mixing time underestimation leads to overconfident bounds; our conservative strategy of taking the maximum over multiple autocorrelation sources trades some tightness for reliability. The KL divergence between Gaussian posteriors, while analytically convenient, does not respect the geometry of the parameter space (Viallard et al., 2023) and may limit posterior expressiveness. Future work could explore Wasserstein-based alternatives (Amit et al., 2022; Haddouche & Guedj, 2023; Viallard et al., 2023), structured posteriors, adaptive mixing time estimation, and extensions beyond actor-critic architectures. Despite these constraints, our results establish PAC-Bayesian computation as a viable path to trustworthy deep RL (Xu et al., 2022).

## Acknowledgements

We gratefully acknowledge support from the CNRS/IN2P3 Computing Center (Lyon - France) for providing computing resources needed for this work.

## Impact Statement

This paper presents work whose goal is to advance the field of machine learning, specifically reinforcement learning. Our primary contributions are motivated by the quest for performance certificates for deep RL to pave the way for future safer deployment in high-stakes applications such as robotics and autonomous systems. There are many potential societal consequences of our work, none of which we feel must be specifically highlighted here.

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

# A. Mathematical Tools

**Lemma A.1** (Markov's Inequality). *For any random variable $X$, for any $a > 0$, we have*

$$\mathbb{P}\{|X| \geq a\} \leq \frac{\mathbb{E}[|X|]}{a}.$$

**Lemma A.2** (Change of measure inequality). *For any measurable function $f : \Theta \to \mathbb{R}$ and distributions $\mu, \rho \in \mathcal{P}(\Theta)$:*

$$\mathbb{E}_{\theta \sim \rho}[f(\theta)] \leq \mathrm{KL}(\rho\|\mu) + \ln \mathbb{E}_{\theta \sim \mu}[\exp(f(\theta))] \tag{15}$$

*where $\mathrm{KL}(\rho\|\mu)$ is the Kullback-Leibler divergence.*

## A.1. Concentration for Markov chains via Marton coupling

We use Paulin's extension of McDiarmid's bounded-difference inequality to Markov chains (Paulin, 2018). This extension provides concentration inequalities for functions of dependent random variables, with constants that depend on the mixing properties of the chain.

### A.1.1. MARTON COUPLING AND MIXING TIME

The key insight in Paulin's approach (Paulin, 2018) is to use a coupling structure known as Marton coupling, which quantifies the strength of dependence between random variables in a discrete-time stochastic process, particularly for Markov chains.

Consider a vector of (generally dependent) random variables $X = (X_1, \ldots, X_N)$ taking values in space $\Lambda = \Lambda_1 \times \ldots \times \Lambda_N$. A Marton coupling provides a way to couple the distributions of future states $X_{i+t}$ conditioned on different past states $X_i$. Let $\mathcal{L}(X_{i+t}|X_i = x)$ be the conditional law (conditional distribution) of $X_{i+t}$ given $X_i = x$.

For a Markov chain $X = (X_1, \ldots, X_N)$, let $\tau(\varepsilon)$ denote the mixing time of the chain in total variation distance, defined as the minimal $t$ such that for every $1 \leq i \leq N - t$ and $x, y \in \Lambda_i$ the following holds:

$$d_{\mathrm{TV}}(\mathcal{L}(X_{i+t}|X_i = x), \mathcal{L}(X_{i+t}|X_i = y)) \leq \varepsilon \tag{16}$$

We define the normalized mixing time parameter $\tau_{\min}$ as:

$$\tau_{\min} = \inf_{0 \leq \varepsilon < 1} \tau(\varepsilon) \left( \frac{2 - \varepsilon}{1 - \varepsilon} \right)^2 \tag{17}$$

### A.1.2. MCDIARMID'S INEQUALITY FOR MARKOV CHAINS

Consider a function $f : \Lambda \to \mathbb{R}$ satisfying the bounded-differences property: For any $x, y \in \Lambda$,

$$f(x) - f(y) \leq \sum_{i=1}^{N} c_i \mathbb{I}[x_i \neq y_i] \tag{18}$$

where $c \in \mathbb{R}_+^N$ and $\mathbb{I}[\text{condition}]$ is the indicator function (equal to 1/0 according to whether the 'condition' is true/false). Paulin's theorem then gives:

$$\Pr(|f(X) - \mathbb{E}f(X)| \geq t) \leq 2 \exp(-2t^2/\|c\|^2 \tau_{\min}). \tag{19}$$

The squared norm $\|c\|^2$ on the right-hand side is the squared Euclidean norm, defined as usual: $\|c\|^2 = \sum_{i=1}^{N} c_i^2$.

### A.1.3. APPLICATION TO BOUNDED DIFFERENCES IN MDPs

For Markov decision processes, this inequality is particularly useful when analyzing the difference between value functions. If perturbing a single transition can change the value by at most $c_i$, then the total effect on a function of trajectories satisfies the bounded-differences property, and then the deviation from its mean can be bounded by the above concentration inequality, with the mixing time of the MDP properly accounting for the propagation of the perturbation through future states.

# B. Derivation of PAC-Bayes Value-Error Bound for RL

## B.1. Bounded-differences property for MDP trajectories

We begin by recalling the definitions of discounted return for a trajectory $\xi$ and the corresponding value function from Section 3:

$$G(\xi) = \sum_{k=0}^{H-1} \gamma^k R_{k+1}$$

$$V_{\pi_\theta} = \mathbb{E}_{\xi \sim \mathcal{M}}[G(\xi)]$$

As defined in (8) and (9), the empirical and expected losses are:

$$\hat{\mathcal{L}}_{\mathfrak{D}}(\theta) = -\frac{1}{T} \sum_{j=1}^{T} G(\xi^{(j)})$$

$$\mathcal{L}(\theta) = -\mathbb{E}_{\xi \sim \mathcal{M}}[G(\xi)] = -V_{\pi_\theta}$$

To apply Paulin's inequality, we shall establish the bounded-differences condition for this empirical loss. Specifically, if we show that replacing one transition in a trajectory affects $\hat{\mathcal{L}}_{\mathfrak{D}}(\theta)$ by at most $c_{(h,j)}$, then this will give a bounded-differences property of the form

$$\hat{\mathcal{L}}_{\mathfrak{D}}(\theta) - \hat{\mathcal{L}}_{\bar{\mathfrak{D}}}(\theta) \le \sum_{h=1}^{H} \sum_{j=1}^{T} c_{(h,j)} \mathbb{I}[\xi_h^{(j)} \ne \bar{\xi}_h^{(j)}]$$

where $c \in \mathbb{R}_+^{H \times T}$ and $\mathbb{I}[\cdot]$ is the indicator function. This way, Paulin's inequality (19) applied to this setting then would give the probability tail inequality establishing the concentration of $\hat{\mathcal{L}}_{\mathfrak{D}}(\theta)$ around its mean $\mathcal{L}(\theta)$.

## B.2. Quantifying the impact of perturbed transitions

Suppose we replace a single transition at position $h$ in trajectory $j$. This perturbation alters the trajectory $\xi^{(j)}$ to $\bar{\xi}^{(j)}$ and effectively changes the cumulative importance weight associated with that trajectory.

By our assumption in Section 3.1, the weighted rewards are effectively bounded in $[0, R_{\max}]$. Therefore, we can bound the sensitivity of the empirical loss by treating the weighted signal similarly to the on-policy case. The maximum change in the weighted discounted return due to a perturbation at step $h$ is bounded by the propagation of the error through the discount factor:

$$|\hat{\mathcal{L}}_{\mathfrak{D}}(\theta) - \hat{\mathcal{L}}_{\bar{\mathfrak{D}}}(\theta)| \le \frac{\gamma^{h-1} R_{\max}}{T} = c_{(h,j)}$$

**Validity under weight clipping.** Clipping large importance weights for numerical stability introduces bias into the empirical loss. We now verify that this bias is strictly pessimistic and leaves the certificate intact.

Let the clipped weights be $w_j^c = \min(w_j, M)$ for some threshold $M > 0$ and define the clipped empirical and true losses:

$$\hat{\mathcal{L}}_{\mathfrak{D}}^c(\theta) = -\frac{1}{T} \sum_{j=1}^{T} w_j^c \, G(\xi^{(j)}), \qquad \mathcal{L}_c(\theta) = -\mathbb{E}_{\xi \sim \mathcal{M}}[w^c \, G(\xi)].$$

Since $G(\xi^{(j)}) \ge 0$ and $w_j^c \le w_j$, we have $w_j^c G(\xi^{(j)}) \le w_j G(\xi^{(j)})$ for every trajectory, hence

$$\mathcal{L}(\theta) \le \mathcal{L}_c(\theta). \tag{20}$$

Applying Theorem 3.3 to the clipped loss $\mathcal{L}_c$ (which satisfies the same bounded-differences condition since $w_j^c G \le w_j G \le R_{\max}$ pointwise) gives, with probability at least $1 - \delta$:

$$\mathcal{L}_c(\theta) \le \hat{\mathcal{L}}_{\mathfrak{D}}^c(\theta) + \sqrt{\frac{R_{\max}^2 (1 - \gamma^{2H})}{T(1 - \gamma^2)} \tau_{\min} \left( \mathrm{KL}(\rho \| \mu) + \ln \frac{\sqrt{2}}{\delta} \right)}.$$

Chaining with (20):

$$\mathcal{L}(\theta) \; \leq \; \hat{\mathcal{L}}_{\mathfrak{D}}^{c}(\theta) + \text{complexity term.}$$

Hence the certificate remains formally valid under clipping, it simply becomes more conservative. In our implementation we use Softmax Self-Normalised Importance Sampling over standardised log-ratios, which naturally constrains all weights to $[0, 1]$ and enjoys the same conservative property (see Remark 3.1 in the main text).

### B.3. Derivation of $\|c\|^2$ for the PAC-Bayes bound

To apply McDiarmid's inequality for Markov chains as developed by Paulin (2018), we need to compute $\|c\|^2$:

$$\|c\|^2 = \sum_{j=1}^{T} \sum_{h=1}^{H} c^2(h, j)$$

$$= \frac{R_{\max}^2}{T^2} \cdot T \sum_{h=1}^{H} \gamma^{2(h-1)}$$

$$= \frac{R_{\max}^2}{T} \underbrace{\sum_{h=0}^{H-1} \gamma^{2h}}_{\text{finite geometric series}}$$

$$= \frac{R_{\max}^2}{T} \cdot \frac{1 - \gamma^{2H}}{1 - \gamma^2} .$$

For infinite-horizon settings where $H \to \infty$ and $\gamma < 1$, the series converges to $1/(1 - \gamma^2)$, this simplifies to

$$\|c\|^2 = \frac{R_{\max}^2}{T(1 - \gamma^2)}.$$

### B.4. Full accounting of perturbation propagation effects

A critical question is whether our derivation of $\|c\|^2$ fully accounts for the propagation of perturbations through the trajectory. For likelihood ratio divergence, we rely on the standard assumption (Section 3.1 and Remark 3.1) that weighted rewards remain bounded in $[0, R_{\max}]$, so perturbations to the importance-weighted signal are contained within the same range as on-policy rewards. Since a perturbation at step $h$ in trajectory $j$ affects all subsequent transitions in that trajectory, the bounded-differences indicator is 1 for every $(h', j)$ with $h' \geq h$.

For a perturbation at step $h$ in trajectory $j$, the sum of corresponding coefficients is:

$$\sum_{h'=h}^{H} c_{(h',j)} = \frac{R_{\max}}{T} \sum_{k=0}^{H-h} \gamma^{h-1+k} = \frac{R_{\max}\gamma^{h-1}}{T} \cdot \frac{1 - \gamma^{H-h+1}}{1 - \gamma}$$

The actual maximum change in discounted return from this perturbation (worst case: reward changes from 0 to $R_{\max}$) is:

$$|G(\xi^{(j)}) - G(\bar{\xi}^{(j)})| \leq R_{\max}\gamma^{h-1} \sum_{k=0}^{H-h} \gamma^k = R_{\max}\gamma^{h-1} \cdot \frac{1 - \gamma^{H-h+1}}{1 - \gamma}$$

When divided by $T$ (because $\hat{\mathcal{L}}_{\mathfrak{D}}(\theta)$ averages over $T$ trajectories), we get exactly the same quantity as the sum of coefficients above. Therefore, the bounded-differences condition holds with equality, confirming that our derivation of $\|c\|^2$ fully accounts for all propagation effects.

This careful accounting of propagation effects allows us to apply McDiarmid's inequality for Markov chains (Paulin, 2018) to obtain the PAC-Bayes bound in Theorem 3.3 with the correct constants.

## B.5. Derivation of the PAC-Bayes Bound

Having established the bounded-differences property and quantified the impact of perturbations via $\|c\|^2$, we now proceed to derive the PAC-Bayes bound on the expected difference between empirical and true losses.

### B.5.1. FROM MCDIARMID TO MOMENT GENERATING FUNCTION

McDiarmid's inequality for Markov chains (inequality (19), Paulin, 2018) provides a concentration inequality in the form of a probability tail inequality for the deviation between empirical and expected losses. From this, via well-known techniques we can derive a bound on the moment generating function (MGF):

**Lemma B.1** (MGF bound for Markov chains). *For any $\kappa > 0$ and policy parameters $\theta \in \Theta$:*

$$\mathbb{E}_{\mathfrak{D} \sim \mathcal{M}^{(T)}} \left[ \exp\left( \kappa(\mathcal{L}(\theta) - \hat{\mathcal{L}}_{\mathfrak{D}}(\theta)) \right) \right] \leq \exp\left( \frac{\kappa^2 \|c\|^2 \tau_{\min}}{8} \right) \tag{21}$$

*where $\tau_{\min}$ is the mixing time of the Markov chain induced by a behavior policy $\pi_b$.*

### B.5.2. FINAL BOUND VIA SUB-GAUSSIAN QUADRATIC MOMENT

A direct application of Lemma B.1 via the PAC-Bayesian change of measure (Lemma A.2) yields a bound parametrized by the free variable $\kappa > 0$: for any fixed posterior $\rho$ and any $\kappa > 0$,

$$\mathbb{E}_{\mathfrak{D}} \mathbb{E}_{\theta \sim \rho}[\mathcal{L}(\theta) - \hat{\mathcal{L}}_{\mathfrak{D}}(\theta)] \leq \frac{\mathbb{E}_{\mathfrak{D}} \left[ \mathrm{KL}(\rho \| \mu) \right]}{\kappa} + \frac{\kappa \|c\|^2 \tau_{\min}}{8} \tag{22}$$

This could then be used to obtain a high-probability bound, by using Markov's inequality. Unfortunately, said bound would hold for a fixed $\kappa$, preventing minimization over $\kappa$. A high-probability bound that holds uniformly over $\kappa$ could be obtained via a grid over $\kappa$ and a union bound, this argument is given in Appendix B.5.4. In this section, instead of optimizing over $\kappa$, we leverage the sub-Gaussian nature of the generalization error to derive a tighter bound directly.

From Lemma B.1 (Equation (21)), we can establish that the generalization error $\Delta(\theta) = \mathcal{L}(\theta) - \hat{\mathcal{L}}_{\mathfrak{D}}(\theta)$ satisfies the sub-Gaussian property:

$$\mathbb{E}_{\mathfrak{D}} \left[ \exp(\kappa \Delta(\theta)) \right] \leq \exp\left( \frac{\kappa^2 \|c\|^2 \tau_{\min}}{8} \right) = \exp\left( \frac{\kappa^2 b^2}{2} \right), \quad \text{where } b^2 = \frac{\|c\|^2 \tau_{\min}}{4}. \tag{23}$$

We utilize the property that centered sub-Gaussian random variables satisfy a bound on their quadratic exponential moment. Specifically, following Wainwright (2019) (Theorem 2.6.IV) and the correction by Hellström & Durisi (2020; 2021), for a $b$-sub-Gaussian variable $X$, and for any $\lambda \in [0, 1)$, we have $\mathbb{E}[\exp(\frac{\lambda X^2}{2b^2})] \leq \frac{1}{\sqrt{1-\lambda}}$. Choosing $\lambda = \frac{1}{2}$ for simplicity:

$$\mathbb{E}_{\mathfrak{D}} \left[ \exp\left( \frac{\Delta(\theta)^2}{4b^2} \right) \right] \leq \sqrt{2}. \tag{24}$$

We now apply the PAC-Bayesian change of measure (Lemma A.2) to the function $f(\theta) = \frac{\Delta(\theta)^2}{4b^2}$:

$$\mathbb{E}_{\theta \sim \rho} \left[ \frac{(\mathcal{L}(\theta) - \hat{\mathcal{L}}_{\mathfrak{D}}(\theta))^2}{4b^2} \right] \leq \mathrm{KL}(\rho \| \mu) + \ln \mathbb{E}_{\theta \sim \mu} \left[ \exp\left( \frac{\Delta(\theta)^2}{4b^2} \right) \right]. \tag{25}$$

Taking the expectation with respect to the data distribution $\mathfrak{D} \sim \mathcal{M}^{(T)}$ and applying Jensen's inequality to the concave function $\ln(\cdot)$ followed by Fubini's theorem to swap expectations in the second term:

$$\mathbb{E}_{\mathfrak{D}} \mathbb{E}_{\theta \sim \rho} \left[ \frac{(\mathcal{L}(\theta) - \hat{\mathcal{L}}_{\mathfrak{D}}(\theta))^2}{4b^2} \right] \leq \mathbb{E}_{\mathfrak{D}}[\mathrm{KL}(\rho \| \mu)] + \ln \mathbb{E}_{\theta \sim \mu} \mathbb{E}_{\mathfrak{D}} \left[ \exp\left( \frac{\Delta(\theta)^2}{4b^2} \right) \right] \tag{26}$$

$$\leq \mathbb{E}_{\mathfrak{D}}[\mathrm{KL}(\rho \| \mu)] + \ln(\sqrt{2}), \quad \text{(using 24)}. \tag{27}$$

Multiplying by $4b^2$ and substituting $b^2 = \frac{\|c\|^2 \tau_{\min}}{4}$:

$$\mathbb{E}_{\mathfrak{D}}\mathbb{E}_{\theta \sim \rho}\left[(\mathcal{L}(\theta) - \hat{\mathcal{L}}_{\mathfrak{D}}(\theta))^2\right] \leq \|c\|^2 \tau_{\min}\left(\mathbb{E}_{\mathfrak{D}}[\mathrm{KL}(\rho\|\mu)] + \ln\sqrt{2}\right). \tag{28}$$

Finally, we apply Jensen's inequality to the left-hand side, using $(\mathbb{E}[X])^2 \leq \mathbb{E}[X^2]$:

$$\left(\mathbb{E}_{\mathfrak{D}}\mathbb{E}_{\theta \sim \rho}\left[\mathcal{L}(\theta) - \hat{\mathcal{L}}_{\mathfrak{D}}(\theta)\right]\right)^2 \leq \mathbb{E}_{\mathfrak{D}}\mathbb{E}_{\theta \sim \rho}\left[(\mathcal{L}(\theta) - \hat{\mathcal{L}}_{\mathfrak{D}}(\theta))^2\right]. \tag{29}$$

Taking the square root gives the final in-expectation bound:

$$\left|\mathbb{E}_{\mathfrak{D}}\mathbb{E}_{\theta \sim \rho}[\mathcal{L}(\theta) - \hat{\mathcal{L}}_{\mathfrak{D}}(\theta)]\right| \leq \sqrt{\|c\|^2 \tau_{\min}\left(\mathbb{E}_{\mathfrak{D}}[\mathrm{KL}(\rho\|\mu)] + \frac{1}{2}\ln 2\right)}. \tag{30}$$

### B.5.3. HIGH-PROBABILITY BOUND

To obtain the high-probability result (Theorem 3.3), we apply Markov's inequality to $\mathbb{E}_{\theta \sim \mu}\left[\exp\left(\frac{\Delta(\theta)^2}{4b^2}\right)\right]$ as a random variable over $\mathfrak{D}$ (recall that $\Delta(\theta) = \mathcal{L}(\theta) - \hat{\mathcal{L}}_{\mathfrak{D}}(\theta)$), whose expectation over $\mathfrak{D}$ is at most $\sqrt{2}$ by (24) and Fubini. This gives a high-probability event that does not involve $\rho$; the change-of-measure inequality (Lemma A.2) then yields the following (similar to Hellström & Durisi (2021, Eq. 3)): with probability at least $1 - \delta$ over $\mathfrak{D}$, simultaneously for all $\rho \in \mathcal{P}(\Theta)$,

$$\mathbb{E}_{\theta \sim \rho}\left[(\mathcal{L}(\theta) - \hat{\mathcal{L}}_{\mathfrak{D}}(\theta))^2\right] \leq 4b^2\left(\mathrm{KL}(\rho\|\mu) + \ln\frac{\sqrt{2}}{\delta}\right). \tag{31}$$

Using Jensen's inequality $(\mathbb{E}_{\theta \sim \rho}[\Delta(\theta)])^2 \leq \mathbb{E}_{\theta \sim \rho}[\Delta(\theta)^2]$ and taking the square root:

$$\left|\mathbb{E}_{\theta \sim \rho}[\mathcal{L}(\theta) - \hat{\mathcal{L}}_{\mathfrak{D}}(\theta)]\right| \leq \sqrt{\|c\|^2 \tau_{\min}\left(\mathrm{KL}(\rho\|\mu) + \ln\frac{\sqrt{2}}{\delta}\right)}. \tag{32}$$

Substituting $\|c\|^2$ from (12) yields the result in Theorem 3.3.

Recalling that $\mathcal{L}(\theta) = -V_{\pi_\theta}$ from (8), we obtain the PAC-Bayes lower bound on the expected value function:

$$\mathbb{E}_{\theta \sim \rho}[V_{\pi_\theta}] \geq \mathbb{E}_{\theta \sim \rho}[-\hat{\mathcal{L}}_{\mathfrak{D}}(\theta)] - \sqrt{\|c\|^2 \tau_{\min}\left(\mathrm{KL}(\rho\|\mu) + \ln\frac{\sqrt{2}}{\delta}\right)}. \tag{33}$$

where the true expected value is lower-bounded by an empirical estimate minus an uncertainty term that accounts for limited data $(1/T)$, temporal correlations $(\tau_{\min})$, and posterior complexity $(\mathrm{KL}(\rho\|\mu))$.

### B.5.4. OPTIMIZATION OF THE BOUND

The high-probability bound derived above takes the following form: simultaneously for all $\rho$, it holds that

$$\mathbb{E}_{\theta \sim \rho}[\mathcal{L}(\theta)] \leq \underbrace{\mathbb{E}_{\theta \sim \rho}[\hat{\mathcal{L}}_{\mathfrak{D}}(\theta)]}_{\text{Linear in } \rho} + \underbrace{\sqrt{\|c\|^2 \tau_{\min}\left(\mathrm{KL}(\rho\|\mu) + \ln\frac{\sqrt{2}}{\delta}\right)}}_{\text{Concave in KL (Non-convex in } \rho)}. \tag{34}$$

Directly optimizing this objective with respect to the posterior $\rho$ is challenging because the square root of the KL divergence makes the overall objective non-convex. To address this, we employ a variational approximation to obtain a surrogate objective that is amenable to alternating convex optimization.

We utilize the algebraic identity for the square root function: for any $x \geq 0$, $\sqrt{x} = \inf_{\lambda > 0} \left( \frac{x}{2\lambda} + \frac{\lambda}{2} \right)$. By identifying $x$ with the complexity term inside the square root, we obtain an upper bound on the true risk that depends on the auxiliary parameter $\lambda$:

$$\mathbb{E}_{\theta \sim \rho}[\mathcal{L}(\theta)] \leq \inf_{\lambda > 0} \mathcal{J}(\rho, \lambda), \tag{35}$$

where the surrogate objective $\mathcal{J}(\rho, \lambda)$ is defined as:

$$\mathcal{J}(\rho, \lambda) = \mathbb{E}_{\theta \sim \rho}[\hat{\mathcal{L}}_{\mathfrak{D}}(\theta)] + \frac{\|c\|^2 \tau_{\min} \left( \mathrm{KL}(\rho \| \mu) + \ln \frac{\sqrt{2}}{\delta} \right)}{2\lambda} + \frac{\lambda}{2}. \tag{36}$$

This surrogate objective has two desirable properties that justify the alternating optimization procedure in our algorithm (Section 4):

**Convexity in $\rho$:** For a fixed $\lambda$, the term becomes a linear scaling of the KL divergence, which is convex in $\rho$. This allows for stable gradient-based updates of the posterior parameters.

**Convexity in $\lambda$:** For a fixed $\rho$, the function is strictly convex in $\lambda$, allowing for an analytic solution:

$$\lambda^* = \sqrt{\|c\|^2 \tau_{\min} \left( \mathrm{KL}(\rho \| \mu) + \ln \frac{\sqrt{2}}{\delta} \right)}. \tag{37}$$

Minimizing $\mathcal{J}(\rho, \lambda)$ via alternating updates is therefore mathematically equivalent to minimizing the upper bound of the rigorous high-probability inequality derived in Section B.5.3.

## C. Alternative derivation via union bound and a grid over $\kappa$

Below, we provide another alternative derivation, keeping the parameter $\kappa$ and optimizing it via a grid search and union bound following the procedure from Alquier (2024).

### C.1. PAC-Bayes change of measure

Starting from the MGF bound above (Lemma B.1), we can follow the standard PAC-Bayes derivation. Let $\Theta$ be our parameter space and let $\mu \in \mathcal{P}(\Theta)$ be a prior distribution over $\Theta$ chosen independently of the data. For any posterior distribution $\rho \in \mathcal{P}(\Theta)$ (which may depend on $\mathfrak{D}$), we apply the change of measure inequality that follows from Donsker–Varadhan variational formula (Donsker & Varadhan, 1975; 1983).

Let $f(\theta) = \kappa(\mathcal{L}(\theta) - \hat{\mathcal{L}}_{\mathfrak{D}}(\theta))$. Applying Lemma A.2:

$$\mathbb{E}_{\theta \sim \rho}[\kappa(\mathcal{L}(\theta) - \hat{\mathcal{L}}_{\mathfrak{D}}(\theta))] \leq \mathrm{KL}(\rho \| \mu) + \ln \mathbb{E}_{\theta \sim \mu}[\exp(\kappa(\mathcal{L}(\theta) - \hat{\mathcal{L}}_{\mathfrak{D}}(\theta)))] \tag{38}$$

### C.2. Combining with the MGF bound

Taking the expectation with respect to $\mathfrak{D} \sim \mathcal{M}^{(T)}$ on both sides:

$$\mathbb{E}_{\mathfrak{D}} \mathbb{E}_{\theta \sim \rho}[\kappa(\mathcal{L}(\theta) - \hat{\mathcal{L}}_{\mathfrak{D}}(\theta))] \leq \mathbb{E}_{\mathfrak{D}}[\mathrm{KL}(\rho \| \mu)] + \mathbb{E}_{\mathfrak{D}}\left[ \ln \mathbb{E}_{\theta \sim \mu}[\exp(\kappa(\mathcal{L}(\theta) - \hat{\mathcal{L}}_{\mathfrak{D}}(\theta)))] \right] \tag{39}$$

Applying Jensen's inequality to the concave function $\ln(\cdot)$ yields:

$$\mathbb{E}_{\mathfrak{D}} \mathbb{E}_{\theta \sim \rho}[\kappa(\mathcal{L}(\theta) - \hat{\mathcal{L}}_{\mathfrak{D}}(\theta))] \leq \mathbb{E}_{\mathfrak{D}}[\mathrm{KL}(\rho \| \mu)] + \ln \mathbb{E}_{\mathfrak{D}}\left[ \mathbb{E}_{\theta \sim \mu}[\exp(\kappa(\mathcal{L}(\theta) - \hat{\mathcal{L}}_{\mathfrak{D}}(\theta)))] \right] \tag{40}$$

By Fubini-Tonelli's theorem, we swap the order of expectations in the second term and apply the MGF bound from Lemma B.1:

$$\mathbb{E}_{\mathfrak{D}} \mathbb{E}_{\theta \sim \rho}[\kappa(\mathcal{L}(\theta) - \hat{\mathcal{L}}_{\mathfrak{D}}(\theta))] \leq \mathbb{E}_{\mathfrak{D}}[\mathrm{KL}(\rho \| \mu)] + \ln \mathbb{E}_{\theta \sim \mu} \mathbb{E}_{\mathfrak{D}}[\exp(\kappa(\mathcal{L}(\theta) - \hat{\mathcal{L}}_{\mathfrak{D}}(\theta)))] \tag{41}$$

$$\leq \mathbb{E}_{\mathfrak{D}}[\mathrm{KL}(\rho \| \mu)] + \ln \mathbb{E}_{\theta \sim \mu}\left[ \exp\left( \frac{\kappa^2 \|c\|^2 \tau_{\min}}{8} \right) \right] \tag{42}$$

$$= \mathbb{E}_{\mathfrak{D}}[\mathrm{KL}(\rho \| \mu)] + \frac{\kappa^2 \|c\|^2 \tau_{\min}}{8} \tag{43}$$

Dividing by $\kappa > 0$:

$$\mathbb{E}_{\mathfrak{D}}\mathbb{E}_{\theta \sim \rho}[\mathcal{L}(\theta) - \hat{\mathcal{L}}_{\mathfrak{D}}(\theta)] \leq \frac{\mathbb{E}_{\mathfrak{D}}\left[\mathrm{KL}(\rho\|\mu)\right]}{\kappa} + \frac{\kappa\|c\|^2\tau_{\min}}{8} \tag{44}$$

where The true expected value is lower-bounded by an empirical estimate minus an uncertainty term that accounts for limited data ($1/T$), temporal correlations ($\tau_{\min}$), and posterior complexity ($\mathrm{KL}(\rho\|\mu)$).

To convert (44) into a high-probability bound uniform over a finite grid $\mathbb{K} \subset (0, +\infty)$ of candidate values of $\kappa$, one applies a union bound across all $\kappa \in \mathbb{K}$, at the cost of replacing $\log(1/\delta)$ with $\log(\mathrm{card}(\mathbb{K})/\delta)$ in the final expression. The complete argument—including the precise grid construction and the resulting corollary—is given in Alquier (2024, Section 2.1.4, Theorem 2.4 and Corollary 2.5).

## D. A note on the Markov assumption for Bellman errors

The work of Tasdighi et al. (2025) makes an interesting theoretical contribution by modeling the sequence of Bellman errors as a Markov chain. While this approach provides valuable insights, it is worth examining the conditions under which this assumption holds.

We present a simple illustrative example that highlights when the Markov property may not apply to Bellman error sequences. Consider a basic MDP with four states $\{A, B, C, D\}$ and the following transition dynamics with discount factor $\gamma = 0$:

- State $A$ transitions to state $C$ with reward $r = 0$

- State $B$ transitions to state $D$ with reward $r = 0$

- State $C$ has a self-loop with reward $r = +1$

- State $D$ has a self-loop with reward $r = -1$

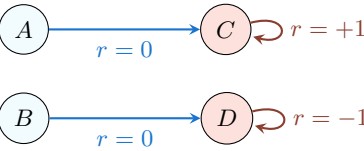

*Figure 3.* A basic four-state MDP

Using a value function $V \equiv 0$ that assigns zero value to all states, we can compute the Bellman errors:

$$\delta_t(A) = r(A) + \gamma \max_a \mathbb{E}[V(s')|s = A, a] - V(A) = 0 + 0 \cdot V(C) - 0 = 0 \tag{45}$$

$$\delta_t(B) = r(B) + \gamma \max_a \mathbb{E}[V(s')|s = B, a] - V(B) = 0 + 0 \cdot V(D) - 0 = 0 \tag{46}$$

Both states $A$ and $B$ yield the same Bellman error $\delta_t = 0$ at time $t$. However, the subsequent errors differ:

$$\delta_{t+1}(C) = r(C) + \gamma \max_a \mathbb{E}[V(s')|s = C, a] - V(C) = +1 + 0 \cdot V(C) - 0 = +1 \tag{47}$$

$$\delta_{t+1}(D) = r(D) + \gamma \max_a \mathbb{E}[V(s')|s = D, a] - V(D) = -1 + 0 \cdot V(D) - 0 = -1 \tag{48}$$

This example demonstrates that the current Bellman error value $\delta_t = 0$ alone does not uniquely determine the distribution of $\delta_{t+1}$, which depends on the underlying state that generated the current error. Although this example used discount $\gamma = 0$ for the sake of simplicity, it is worth noting that the same example can be done with $\gamma > 0$, leading to the same conclusion.

This observation suggests that the Markov assumption for Bellman error sequences may require additional conditions or refinements to hold. Such considerations could be valuable for future theoretical developments building upon the framework proposed by Tasdighi et al. (2025).

# E. REINFORCE Trick for Policy-Level Gradients

We prove that the REINFORCE trick can be extended from action-level to policy-level gradients, enabling tractable optimization of expectations over policy parameters.

**Theorem E.1** (Policy-Level REINFORCE). *For a continuous posterior distribution $\rho_{v,\sigma}(\theta) = \mathcal{N}(v, \text{diag}(\sigma^2))$ over policy parameters $\theta$, the gradient of the expected return can be computed as:*

$$\nabla_{v,\sigma} \mathbb{E}_{\theta \sim \rho} \left[ \mathbb{E}[R_t | \pi_\theta] \right] = \mathbb{E}_{\theta \sim \rho} \left[ \nabla_{v,\sigma} \log \rho_{v,\sigma}(\theta) \cdot \mathbb{E}[R_t | \pi_\theta] \right] \tag{49}$$

*Proof.* Let $J(\theta) = \mathbb{E}[R_t | \pi_\theta]$ denote the expected return of a specific policy parameterization $\theta$. We express the objective as an expectation over the continuous posterior density $\rho_{v,\sigma}(\theta)$ and apply the log-derivative trick directly:

$$\nabla_{v,\sigma} \mathbb{E}_{\theta \sim \rho} \left[ J(\theta) \right] = \nabla_{v,\sigma} \int_\Theta \rho_{v,\sigma}(\theta) J(\theta) d\theta \tag{50}$$

$$= \int_\Theta \nabla_{v,\sigma} \rho_{v,\sigma}(\theta) J(\theta) d\theta \tag{51}$$

$$= \int_\Theta \rho_{v,\sigma}(\theta) \frac{\nabla_{v,\sigma} \rho_{v,\sigma}(\theta)}{\rho_{v,\sigma}(\theta)} J(\theta) d\theta \quad \text{(Log-derivative identity)} \tag{52}$$

$$= \int_\Theta \rho_{v,\sigma}(\theta) \nabla_{v,\sigma} \log \rho_{v,\sigma}(\theta) J(\theta) d\theta \tag{53}$$

$$= \mathbb{E}_{\theta \sim \rho} \left[ \nabla_{v,\sigma} \log \rho_{v,\sigma}(\theta) \cdot J(\theta) \right] \tag{54}$$

Substituting $J(\theta)$ back into the expression yields the theorem statement. ∎

This result enables us to estimate the gradient via sampling: we sample policies $\{\theta_i\}$ from the posterior $\rho$, evaluate their expected returns, and compute the weighted gradient of the log-probability density. This extends the classical REINFORCE trick from action spaces to parameter spaces, allowing efficient optimization of posterior distributions over policies.

# F. PB-SAC Algorithm

This appendix presents the complete algorithmic details of our PAC-Bayes Soft Actor-Critic (PB-SAC) method. Figure 4 provides a high-level illustration of the algorithm flow described in Section 4, showing the interaction between the actor, critic, and bound computation components. The periodic posterior update (step 2) guides exploration through PAC-Bayes bounds (step 3) while maintaining standard SAC training (step 1) using an adaptive sampling (step 4). Section F.1 provides the complete pseudocode with implementation details, including the posterior-guided exploration mechanism, the alternating bound optimization, and the integration with the critic adaptation loop.

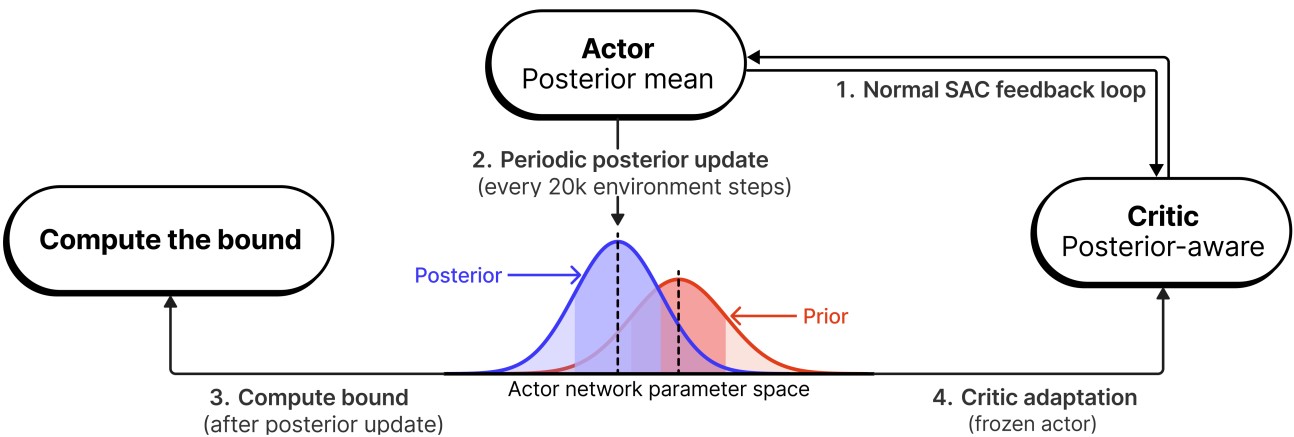

*Figure 4.* Illustration of our algorithm PB-SAC

## F.1. Pseudo Code

---

**Algorithm 1:** PAC-Bayes Soft Actor-Critic (PB-SAC)

---

**Result:** Policy $\pi_\theta$, posterior $\rho$, PAC-Bayes bound

**Init:** Actor $\pi_\theta$, Critics $Q_1, Q_2$, replay buffer $\mathfrak{D}$

**Init:** Posterior $\rho(\theta) = \mathcal{N}(\upsilon, \mathrm{diag}(\sigma^2))$ where $\upsilon$ are the initial actor parameters; Prior $\mu(\theta) = \mathcal{N}(\upsilon_0, \mathrm{diag}(\sigma_0^2))$

**Init:** actor_frozen = False, prior moving average decay $\iota = 0.99$, post. sampling rate $\epsilon_{\mathrm{explore}} = 0.9$ (linearly decayed)

1   **for** $t = 1, 2, \ldots$ **do**

2     **if** *not actor_frozen* **then**

3       `/* `**`Standard SAC + Posterior-Guided Exploration`**` */`

4       **if** *random*$() < \epsilon_{\mathrm{explore}}$ **then**

         `/* Select policy maximizing Q-value from posterior */`

5          Sample multiple policies $\{\theta_i\} \sim \rho$

6          $\theta_{\mathrm{explore}} \leftarrow \arg\max_{\theta_i} Q(s, \pi_\theta(s))$

7          $a \leftarrow \pi_{\theta_{\mathrm{explore}}}(s)$

8       **else**

9          $a \leftarrow \pi_\theta(s)$            `// Current policy (posterior mean)`

10       **end**

11       Execute action $a$ and store transition in $\mathfrak{D}$

12       **if** $|\mathfrak{D}| \geq$ *batch_size* **then**

13          Update critics $Q_1, Q_2$ with standard SAC loss

14          Update actor $\pi_\theta$ with standard SAC loss

15          $\upsilon \leftarrow \theta$            `// Sync posterior mean to current actor`

16       **end**

17     **else**

18       `/* `**`Critic Adaptation Phase (post PAC-Bayes update)`**` */`

19       Sample multiple policies $\{\theta_i\} \sim \rho$ with high sampling rate

20       Compute critic targets averaged over policy samples $\{\theta_i\}$

21       Update critics $Q_1, Q_2$ using averaged targets

       **if:** adaptation steps completed **then** actor_frozen $\leftarrow$ False

22     **end**

23     `/* `**`PAC-Bayes Update Cycle`**` */`

24     **if** $t \bmod$ *pb_update_freq* $= 0$ **then**

25       $\mathfrak{D}_{\mathrm{rollouts}} \leftarrow$ collect_fresh_rollouts()       `// With the current policy`

26       $\tau_{\min} \leftarrow$ estimate_mixing_time($\mathfrak{D}_{\mathrm{rollouts}}$)

27       $\mathfrak{D}_{\mathrm{train}}, \mathfrak{D}_{\mathrm{test}} \leftarrow$ split($\mathfrak{D}_{\mathrm{rollouts}}$)

28       Compute discounted returns $G_{\mathrm{IS}}$ with importance sampling on $\mathfrak{D}_{\mathrm{train}}$

       `/* Alternating optimization */`

29       **for** *epoch* $= 1, \ldots,$ *pb_epochs* **do**

         `/* Optimize posterior parameters for fixed λ */`

30          $\sigma, \upsilon \leftarrow \arg\min_{(\boldsymbol{\sigma}, \boldsymbol{\upsilon})} \mathcal{J}(\rho, \lambda)$       `// inequality 14`

         `/* Optimize λ for fixed posterior */`

31          $\lambda \leftarrow \arg\min_{\boldsymbol{\lambda'}} \mathcal{J}(\rho, \lambda')$       `// inequality 14`

32       **end**

33       bound $\leftarrow$ compute_pac_bayes_bound($\mathfrak{D}_{\mathrm{test}}, \mathrm{KL}(\rho\|\mu), \tau_{\min}$)

34       load_policy_params($\upsilon$)       `// Sync actor to posterior mean`

35       actor_frozen $\leftarrow$ True       `// Initiate critic adaptation`

36     **end**

37     `/* `**`Prior Reset for Maintained Exploration`**` */`

38     **if** $t \bmod$ *pb_reset_freq* $= 0$ **then**

39       $\upsilon_0 \leftarrow \iota \cdot \upsilon + (1 - \iota) \cdot \upsilon_0; \quad \sigma_0 \leftarrow \iota \cdot \sigma + (1 - \iota) \cdot \sigma_0$       `// Moving average prior update`

40       Linearly decay $\iota$ and $\epsilon_{\mathrm{explore}}$

41     **end**

42   **end**

43   **return** $\pi_\theta$, $\rho$, *bound*

---

# G. More Results

This appendix presents extended experimental results that complement the main paper findings. Figure 5 shows detailed performance comparisons on Hopper-v5 and Walker2d-v5 environments, demonstrating both the algorithm comparisons (panel a) and the PAC-Bayes analysis (panel b). The empirical discounted return tracks closely with our certified lower bound, validating our analysis above.

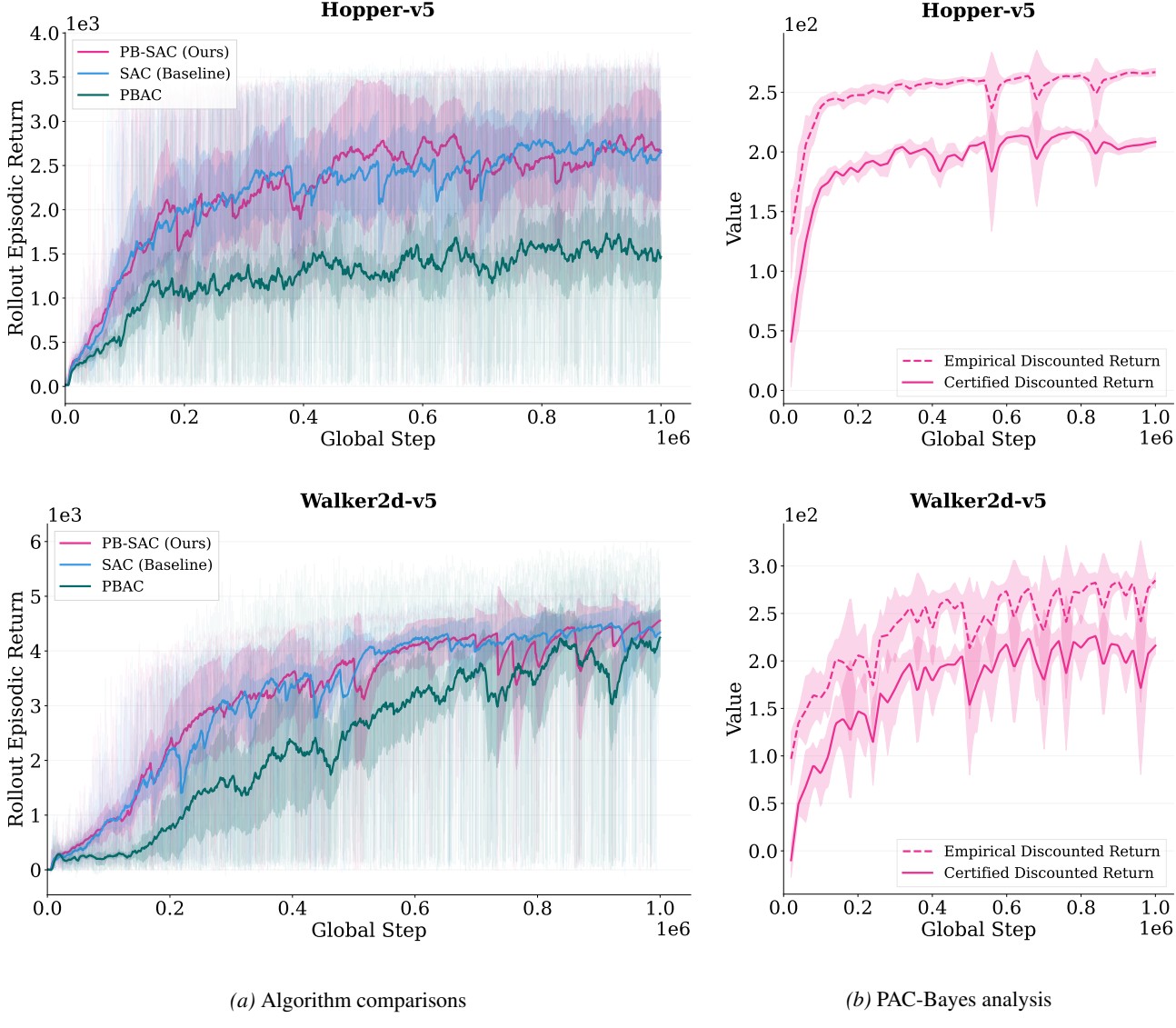

*(a)* Algorithm comparisons

*(b)* PAC-Bayes analysis

*Figure 5.* **(a)** Performance comparison between our ■ PB-SAC, its baseline ■ SAC, and ■ PBAC from Tasdighi et al. (2025); **(b)** PAC-Bayes analysis of PB-SAC across environments. The empirical discounted return (dashed line) corresponds to $\mathbb{E}_{\theta \sim \rho}[-\hat{\mathcal{L}}_{\mathfrak{D}}(\theta)]$, and the certified discounted return (solid line) corresponds to the lower bound on $\mathbb{E}_{\theta \sim \rho}[-\mathcal{L}(\theta)]$ provided by Theorem 3.3 (after rearranging the terms).

## G.1. Sparse-reward setting

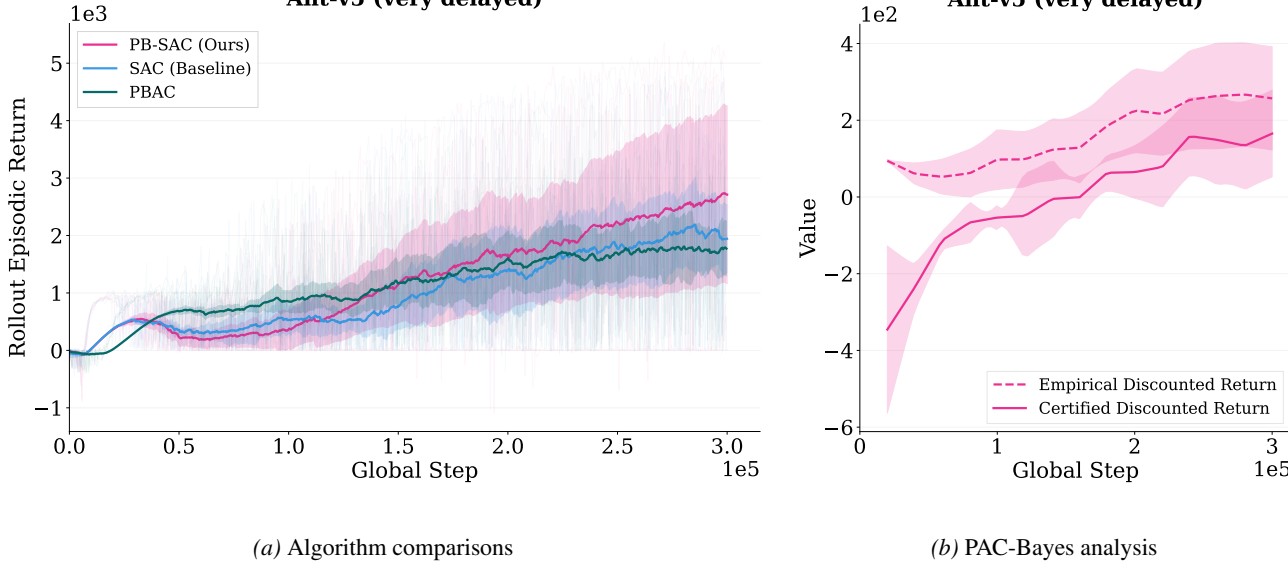

*(a) Algorithm comparisons*

*(b) PAC-Bayes analysis*

*Figure 6.* **(a)** Performance comparison between our ■ PB-SAC, its baseline ■ SAC, and ■ PBAC from Tasdighi et al. (2025); **(b)** PAC-Bayes analysis of PB-SAC across environments. The empirical discounted return (dashed line) corresponds to $\mathbb{E}_{\theta \sim \rho}[-\hat{\mathcal{L}}_{\mathfrak{D}}(\theta)]$, and the certified discounted return (solid line) corresponds to the lower bound on $\mathbb{E}_{\theta \sim \rho}[-\mathcal{L}(\theta)]$ provided by Theorem 3.3 (after rearranging the terms).

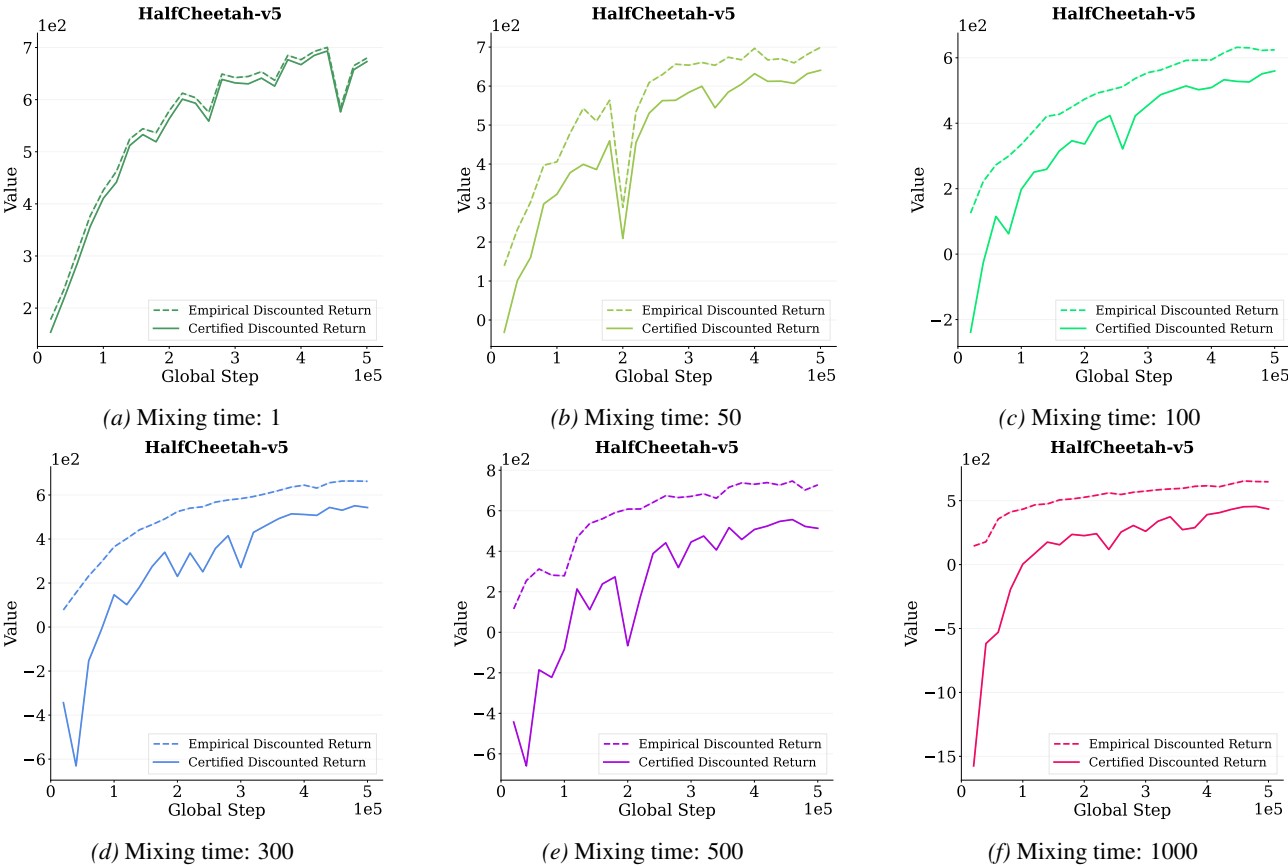

*(a)* Mixing time: 1      *(b)* Mixing time: 50      *(c)* Mixing time: 100

*(d)* Mixing time: 300      *(e)* Mixing time: 500      *(f)* Mixing time: 1000

*Figure 7.* PAC-Bayesian bound behaviour with different fixed mixing time estimates on HalfCheetah-v5. Empirical discounted return (dashed) and certified discounted return (solid) are shown for mixing times ranging from 1 (a) to 1000 (f). Smaller mixing times yield tighter bounds but risk overconfidence if underestimated, while larger values provide conservative but still meaningful certificates. The algorithm maintains practical utility even with substantial mixing time overestimation, demonstrating robustness of the framework.

# H. Ablation study

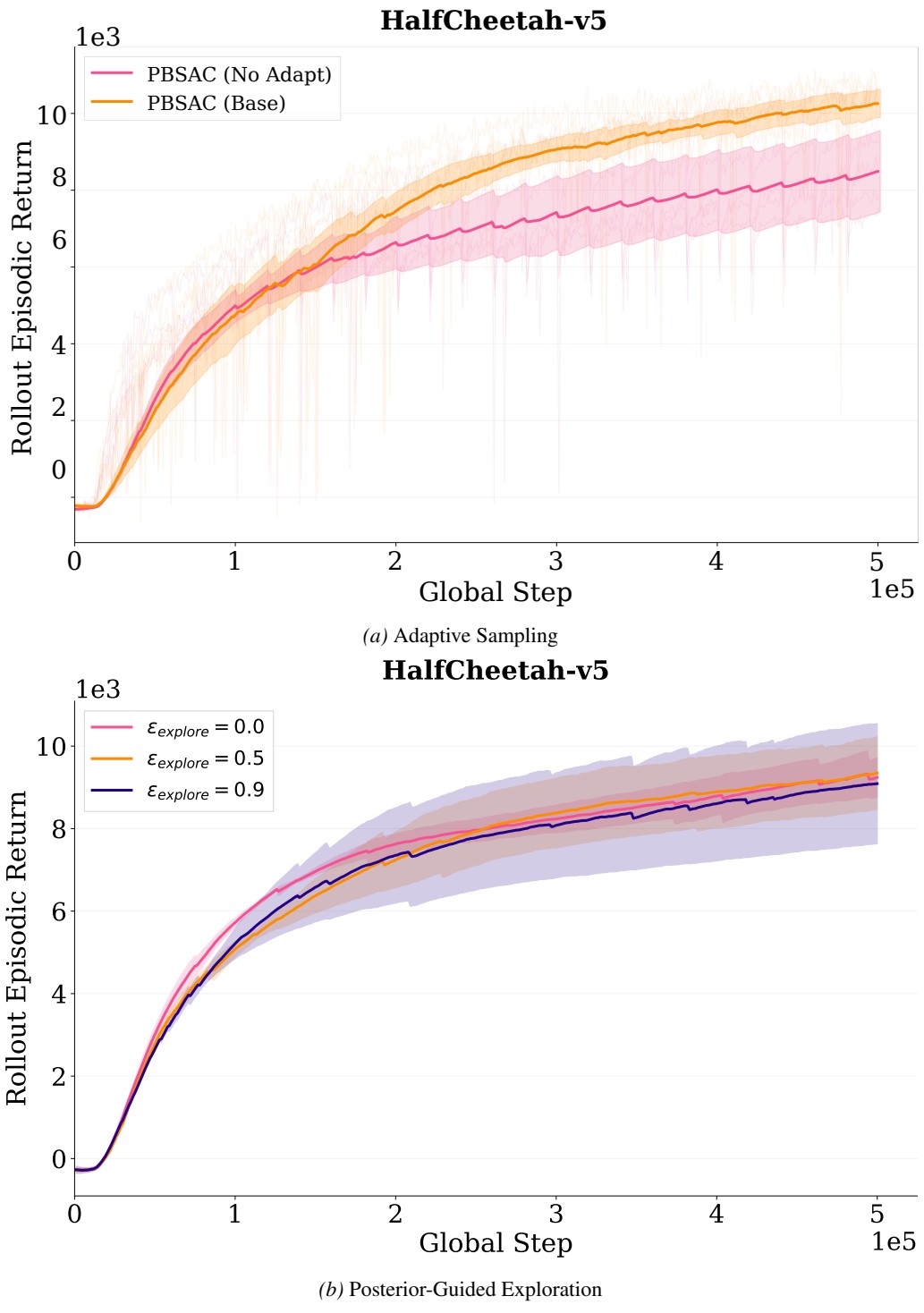

*(a)* Adaptive Sampling

*(b)* Posterior-Guided Exploration

*Figure 8.* Ablations

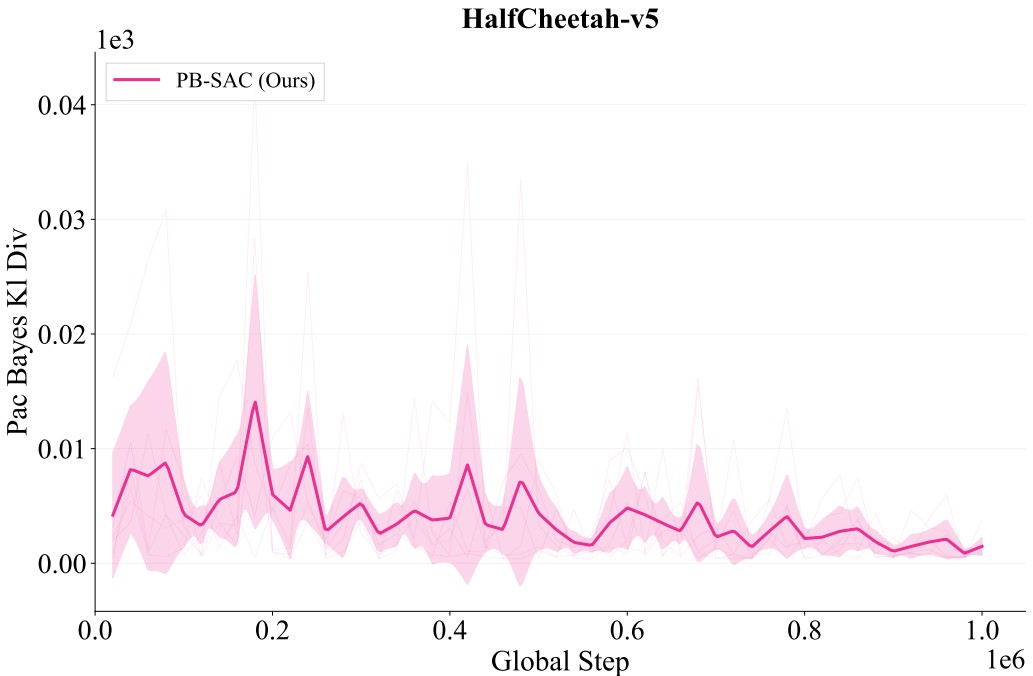

*Figure 9.* KL divergence evolution

## I. Hyperparameter Selection

We carefully selected hyperparameters for our PAC-Bayes Soft Actor-Critic (PB-SAC) implementation to balance performance, sample efficiency, and theoretical guarantees. the common parameters with SAC are left unchanged, while we take the original hyperparameters of PBAC from the paper of Tasdighi et al. (2025). Table 1 summarizes it all.

*Table 1.* Hyperparameter Comparison for MuJoCo Continuous Control Tasks

| Hyperparameter | PB-SAC (Our Algorithm) | SAC (Baseline) | PBAC |
|---|---|---|---|
| **Common SAC Parameters** | | | |
| Total Timesteps | $1 \times 10^6$ | $1 \times 10^6$ | $1 \times 10^6$ |
| Discount Factor ($\gamma$) | 0.99 | 0.99 | 0.99 |
| Soft Update Coefficient ($\tau$) | 0.005 | 0.005 | 0.005 |
| Batch Size | 256 | 256 | 256 |
| Replay Buffer Size | $1 \times 10^6$ | $1 \times 10^6$ | $1 \times 10^6$ |
| Initial Temperature ($\alpha$) | 0.2 | 0.2 | 0.2 |
| Temperature Learning Rate | $3 \times 10^{-4}$ | $3 \times 10^{-4}$ | $3 \times 10^{-4}$ |
| Target Update Frequency | 1 | 1 | 1 |
| **Algorithm-Specific Parameters** | | | |
| Actor Learning Rate | $3 \times 10^{-4}$ | $3 \times 10^{-4}$ | $3 \times 10^{-4}$ |
| Critic Learning Rate | $1 \times 10^{-3}$ | $1 \times 10^{-3}$ | $3 \times 10^{-4}$ |
| Learning Starts | 5,000 | 5,000 | 10,000 |
| Training Frequency | 2 | 2 | 1 |
| Automatic $\alpha$ Tuning | ✓ | ✓ | × |
| Multi-Head Architecture | × | × | ✓ |
| Ensemble of Critics | × | × | ✓ (10) |
| Number of Heads | 1 | 1 | 10 |
| **Network Architecture** | | | |
| Policy Hidden Layers | [256, 256] | [256, 256] | [256, 256] |
| Q-Function Hidden Layers | [256, 256] | [256, 256] | [256, 256] |
| Activation Function | ReLU | ReLU | CReLU |
| **PAC-Bayes Specific (PB-SAC Only)** | | | |
| Failure Probability ($\delta$) | 0.1 | — | — |
| Initial Std Dev | 0.01 | — | — |
| PB Update Frequency | 20,000 | — | — |
| Actor Freeze Frequency | 20 | — | — |
| Adaptation samples | 256 | — | — |
| PB Rollout Trajectories | 100 | — | — |
| PB Rollout Steps per Trajectory | 500 | — | — |
| **PBAC Specific** | | | |
| Bootstrap Rate | — | — | 0.05 |
| Posterior Sampling Rate | — | — | 5 |
| Prior Scaling | — | — | 5.0 |

