# OpenReview forum: "PAC-Bayesian Reinforcement Learning Trains Generalizable Policies"
_ICML.cc/2026/Conference — ICML 2026 regular_

### Official Review · Reviewer_JQYw · 2026-02-16

**Soundness:** 3
**Presentation:** 1
**Significance:** 3
**Originality:** 3
**Overall Recommendation:** 4
**Confidence:** 3

**Summary:**

This paper studies PAC-Bayesian generalization bounds for reinforcement learning under non-independent and identically distributed (non-i.i.d.) data. Based on this bound, the authors design an actor-critic algorithm called PB-SAC, which guides learning through posterior sampling, transforming the generalization guarantee from a passive post-training evaluation tool into an active component of the learning algorithm.

**Compliance With Llm Reviewing Policy:**

Affirmed.

**Final Justification:**

My concerns have been addressed during the rebuttal phase. I hope that the authors can clarify the following points in the final manuscript:
	1.	Non-independent trajectories in PB-SAC and MuJoCo,
	2.	Markovian policies and the tractability of PGE (Key Question 3),
	3.	The likelihood ratio $w_j(\theta)$ and distribution shift due to the assumption that weighted rewards are bounded within $[0, R_{\max}]$.

**Key Questions For Authors:**

1. In the bound presented in Lemma 3.1, why does the likelihood ratio of the trajectory, $w_j(\theta)$, not appear in the final generalization bound? In other words, the bound seems not to account for the distribution shift between the behavior policy and the evaluation policy.
2. One of the claimed contributions of the paper is handling non-independent trajectories. How is this reflected in the design of the PB-SAC algorithm? Moreover, what exactly do non-independent trajectories mean in the MuJoCo continuous control environments used in the experiments?
3. In the Posterior-Guided Exploration section, the algorithm samples policies from the posterior and selects actions that maximize Q-values under posterior uncertainty. However, in the problem setup, the authors allow maximization over non-Markovian policies. Why is it sufficient to maximize $Q(s, \pi_\theta(s))$ for a given state, instead of considering trajectory-dependent quantities such as $Q(\tau)$?
Additionally, if $\pi$ is a non-Markovian policy, how is $\pi_\theta(s)$ defined and computed? Finally, how tractable is it to find $\theta$ that maximizes $Q(s, \pi_\theta(s))$ over $\theta \in \mathcal{I}$?

**Limitations:**

yes

**Strengths And Weaknesses:**

Strength: This paper proposed  bounded-differences property which allows to apply concentration inequalities to dependent data. And the proposed PAC-Bayes bounds improve the dependence on the effective horizon.

Weakness: The paper would benefit from clearer presentation. In particular, the connection between the generalization bound and the PB-SAC algorithm should be better explained, for example by referencing and discussing the graph in Appendix F. In addition, the purpose of Appendices C and D is unclear, as they are never mentioned in the main paper.

---

> ### Author Rebuttal · Authors · 2026-03-30
>
> We sincerely thank the reviewer for their positive assessment of our work. Bellow, we address their questions in detail.
>
> **1. The Likelihood Ratio $w\_j(\\theta)$ and Distribution Shift**
>
> The bound does account for the distribution shift between the behaviour and evaluation policies; this correction happens inside the empirical loss $\\hat{\\mathcal{L}}\_{\\mathfrak{D}}(\\theta)$, which is computed using the importance weights $w_j(\theta)$ (Equation 8).
>
> The reason $w\_j(\\theta)$ does not explicitly appear in the sensitivity constant $c\_{(h,j)}$ (Equation 11) is due to a deliberate theoretical design choice: as stated in Section 3.1, we assume the weighted rewards are bounded within $[0, R\_{\\max}]$ (See also our response to Reviewer **NZH2**). Therefore, the absolute maximum change resulting from altering a single transition is mathematically bottlenecked by $R\_{\\max}$ regardless of the raw policy ratio. We will add a clarifying sentence below Equation (11) to explicitly explain this point.
>
> **2. Non-Independent Trajectories in PB-SAC and MuJoCo**
>
> We appreciate the opportunity to clarify this crucial distinction. The "non-independence" we address refers strictly to intra-trajectory dependence, not inter-trajectory dependence.
>
> - **In MuJoCo:** The transitions _inside_ a single trajectory are highly dependent because they follow a continuous physics simulation. Traditional bounds that assume i.i.d. transitions break down here. Our bound successfully applies to this setting because, as highlighted in Section 3.3, our bounded-differences property is derived at the _transition-level_ rather than the trajectory-level.
>
> - **In PB-SAC:** This theoretical requirement directly informs our algorithmic design. To compute the PAC-Bayes bound correctly, we cannot use transitions sampled uniformly from the standard replay buffer. Instead, as shown in our pseudocode (Appendix F), PB-SAC periodically collects a separate dataset of _fresh rollouts_ using the current frozen actor network. We evaluate the empirical loss on these intact trajectories exactly as they were collected (without shuffling or sampling individual transitions), thereby perfectly preserving their dependent nature so that our bound remains theoretically valid.
>
> **3. Markovian Policies and Tractability of PGE (Key Question 3)**
>
> There is a slight misunderstanding regarding the policy structure that we will clarify in the text.
>
> - **On the sufficiency of $Q(s, \\pi\_\\theta(s))$ over trajectory-dependent quantities.** In the problem setup (Section 3.1), what is (not necessarily) time-homogeneous is the _Markov chain induced by the policy over the finite horizon_, we never claim that the policy function itself is non-Markovian. We believe this answers the first question and exempts us from answering the second one on how $\\pi\_\\theta(s)$ is defined and computed because that follows SAC exactly.
>
> - **On the tractability of $\\arg\\max\_{\\theta_i \\in \\mathcal{I}} Q(s, \\pi\_\\theta(s))$.** As denoted in Section 4.1, The set $\mathcal{I}$ is a finite sample from $\\rho$, drawn at each exploration step (with some probability $\\epsilon\_{explore}$). The maximisation is therefore not a continuous optimisation over $\\Theta$ but a simple forward-pass evaluation over a small discrete candidate set: for each $\\theta\_i \\in \\mathcal{I}$, compute $\\pi\_{\\theta\_i}(s)$ and evaluate $Q(s, \\pi_\{\\theta\_i}(s))$ using the frozen critic, then take the argmax. This is $O(|\\mathcal{I}|)$ forward passes and is fully tractable in practice. The theoretical connection to PAC-Bayesian uncertainty is that $\\mathcal{I}$ concentrates around the posterior mean while the spread of the distribution governs how diverse the candidates are.
>
> **4. Presentation and Appendices (Weakness 1)**
>
> Due to strict space limitations, references to the Appendix were overly condensed.
>
> - We will explicitly reference the PB-SAC architecture diagram (Appendix F) in the opening paragraph of Section 4 to help readers visualise the algorithm.
>
> - Appendix D is a side note discussing the Markov assumption for Bellman errors proposed in the recent work of Tasdighi et al. (2025), demonstrating via a counter-example why such an assumption cannot hold in its current form. We will add a footnote referencing this discussion in Section 3.3 where we discuss previous PAC-Bayesian RL bounds.
>
> - Appendix C is included for theoretical completeness. It derives the bound by optimising over a finite $\kappa$-grid instead of our main, more sophisticated approach. While this alternative is computationally heavier and less tight, we will add a brief reference to it in Section 3.2.
>
> We thank the reviewer again for their constructive review. We hope these clarifications fully address their questions and that they might consider raising their score!

---

> > ### Author Rebuttal · Reviewer_JQYw · 2026-04-01
> >
> > Thanks for your response, I maintain my positive score.

---

> > > ### Author Response · Authors · 2026-04-05
> > >
> > > We sincerely thank the reviewer for their time and for confirming that all of their questions have been fully resolved. We deeply appreciate their helpful feedback.

---

### Official Review · Reviewer_EyBx · 2026-03-11

**Soundness:** 3
**Presentation:** 3
**Significance:** 2
**Originality:** 3
**Overall Recommendation:** 4
**Confidence:** 3

**Summary:**

This paper has two main contributions. The first is a new PAC-Bayesian generalization bound for reinforcement learning which improves upon previous results both by including a tighter dependence on the effective horizon $(1 - \gamma)^{-1}$, and by explicitly depending on the mixing time of the Markov chain, which makes the bound easier to interpret. The second contribution of this paper is a novel algorithm, PB-SAC, which is an augmented version of soft actor-critic (SAC) in which an explicit posterior distribution over policies is maintained and exploration is performed via a posterior-sampling scheme.

**Compliance With Llm Reviewing Policy:**

Affirmed.

**Final Justification:**

In light of the authors' latest response, I am updating my score to weak accept. I think it represents an interesting contribution.

**Key Questions For Authors:**

1. Can you explain why maximizing (13) would yield a solution that includes "theoretically-justified exploration"?
2. Figure 1: are the y-axes correct in all subplots? In particular, why are the "empirical discounted returns" in the right hand plots an order of magnitude less than the "rollout episodic returns"?
3. In Equation (7), why is the value associated with policy $\pi_\theta$ if the notation $\xi \sim \mathcal M$ denotes a trajectory sampled with policy $\pi_b$?
4. At the beginning of Section 3.1, you mention that the Markov chain $\xi$ is not necessarily time-homogeneous. Does this mean that the policy $\pi_\theta$ is non-stationary? This would make sense if the horizon $H$ is assumed to be finite and fixed.

**Limitations:**

yes

**Strengths And Weaknesses:**

I enjoyed reading this paper, it is very well written and the theoretical result is interesting and, as far as I can tell, novel and relevant. In its current form, I recommend a weak rejection, because of my following concern. If the authors can elaborate on this point, I am happy to increase my score.

My concern is that I don't understand the motivation behind the PB-SAC algorithm. On the surface, it is a very complex method, which augments soft actor-critic, but does not yield improved performance. I am personally not familiar with the history of applications of the PAC-Bayesian framework to reinforcement learning, which I believe could provide this motivation. To put it bluntly, I am wondering _what is the point of the PB-SAC algorithm?_ As it does not improve upon previous algorithms, neither in performance nor in computational or conceptual complexity, it seems to me that this algorithm is more of a "proof of concept" for applications of the PAC-Bayesian framework in RL. As an outsider to this line of research, I would need to understand why this is in an interesting contribution. (What is the promise that PAC-Bayesian methods hold?)

One final small comment: Lemma 3.1, as written now, states that only the transition $\xi_h^{(j)}$ has been changed. For equation (10), this would mean that it says "there exists a number $c$ that is larger than (LHS)". In this form, the statement is not very meaningful. If the statement is actually about changes to an arbitrary number of transitions, and if the form of $c$ given in equation (11) works, then it becomes meaningful.

---

> ### Author Rebuttal · Authors · 2026-03-29
>
> We sincerely thank the reviewer for their encouraging review. We are thrilled that they found the paper well-written and the theoretical results novel and relevant.
>
> ## 1. The Promise of PAC-Bayesian RL and the Point of PB-SAC
>
> It is a very fair question: if PB-SAC doesn't outperform standard SAC, what is the point? The core promise of PAC-Bayesian RL isn't raw score maximisation, but _certified generalisation_. It might be worth highlighting that PB-SAC does not outperform and at the same time it does not do worse than standard SAC, whilst the advantage of PB-SAC is in bringing in certified learning.
>
> The reviewer would agree that in critical and high-stakes applications like robotics or medicine, relying solely on empirical scores is unreliable due to overfitting; safe deployment requires performance certificates on unseen data.
>
> In line with this, Our work specifically contributes to the "Generalization" component of Trustworthy RL (See Figure 1 in  Xu et al. (2022) [1]). A recent comprehensive survey by Xu et al. (2022) [1] highlights this exact gap, asking _"how to define the certification goal (e.g., the lower bound of reward)"_ and _"how to train a certifiably generalizable RL."_ noting that this direction has profound theoretical and practical impacts.
>
> PB-SAC serves as a high-impact proof-of-concept that directly answers this open challenge. We contribute:
>
> - **Theory & Algorithm:** A non-vacuous PAC-Bayesian bound adapted for RL data dependencies, optimised dynamically during training via PB-SAC.
>
> - **Empirical Demonstrations:** Validation across dense-reward tasks (safely matching SAC) and complex sparse-reward tasks (where the bound improves exploration to succeed where SAC fails, per Figure 5 and our response to reviewer **iHmT** regarding weakness 2a).
>
> - **Reusable Infrastructure:** Clean, newly designed, open-source code (See supplemental materials; It will be released after publication) to facilitate further research in RL.
>
> Whilst matching SAC in dense tasks, it uniquely provides the exact "lower bound on reward" requested by the trustworthy RL community. We are happy to update the Introduction to highlight this broader motivation.
>
> ## 2. Lemma 3.1 Wording
>
> We thank the reviewer for catching this. We agree that simply stating "there exists a $c$" is weak. The Lemma is indeed meant to state that for _any_ arbitrary single-transition change, the bounded difference holds for the specific $c$ defined in Equation (11). In the revised version, we will rewrite that part to ensure it is mathematically meaningful.
>
> ## 3. Answers to Questions
>
> - **Q1: about "theoretically-justified exploration"**
>
>     We kindly correct that the upper bound in Equation (13) is to be _minimised_, it is a trade-off between maximising empirical return ($-\\hat{\\mathcal{L}}\_{\\mathfrak{D}}(\\theta)$) and minimising the KL complexity penalty. Optimising this yields a posterior distribution $\\rho$ on the policy parameter space. By sampling from $\\rho$ (rather than injecting arbitrary action noise) we explore using policies mathematically constrained to be high-performing and bounded by our prior. This safely guides exploration toward promising parameter regions (We briefly demonstrate this empirically in Figure 5, where PB-SAC solves a sparse-reward task on which standard SAC underperforms).
>
> - **Q2: Figure 1 y-axes**
>
>     This is due to the discount factor $\\gamma = 0.99$. Whilst the "rollout episodic return" (left) is the full _undiscounted_ 1000-step sum, the "empirical discounted return" (right) applies the discount. Because discounted weights sum to at most $\\frac{1}{1-0.99} = 100$, these returns are naturally an order of magnitude smaller than the undiscounted 1000-step sum of rewards.
>
> - **Q3: Equation (7) notation**
>
>     We thank the reviewer for catching this typo. We will update it to a clearer notation, something like: $V\_{\\pi\_\\theta} = \\mathbb{E}\_{\\xi \\sim \\pi\_\\theta}[G(\\xi)]$.
>
> - **Q4: Does non-time-homogeneous mean the policy is non-stationary?**
>
>     Yes, exactly. In our setting, non-time-homogeneity arises naturally from the implicit timestep index within finite-horizon episodes, where the effective transition dynamics and the policy's distribution over actions depend on the step $h \\in [H]$ of the trajectory.
>
>     Crucially, this generality is not an assumption we arbitrarily impose, but one we inherit directly from our concentration framework: Paulin (2018) explicitly states Corollary 2.11 without requiring time-homogeneity.
>
> We thank the reviewer again for the incredibly constructive feedback. We hope our explanation of the core motivation and the technical clarifications address their concerns.
>
> **References**
>
> [1] Xu, M., Liu, Z., Huang, P., Ding, W., Cen, Z., Li, B., and Zhao, D. (2022). Trustworthy reinforcement learning against intrinsic vulnerabilities: Robustness, safety, and generalizability. _arXiv preprint arXiv:2209.08025_.

---

> > ### Author Rebuttal · Reviewer_EyBx · 2026-04-03
> >
> > I thank the authors for their response. For now, I will keep my score as-is. Could you please expand on "Promise of PAC-Bayesian RL"? What do you mean by "certified generalisation" of PB-SAC?
> > > The reviewer would agree that in critical and high-stakes applications like robotics or medicine, relying solely on empirical scores is unreliable due to overfitting; safe deployment requires performance certificates on unseen data.
> >
> > So are you saying PB-SAC is "safer" than SAC? What does that mean?

---

> > > ### Author Response · Authors · 2026-04-05
> > >
> > > We thank the reviewer for the follow-up questions.
> > >
> > > We understand the reviewer's concern regarding the term "safety," as it carries very specific and sometimes differing meanings across RL/ML subfields.
> > >
> > > When we say "certified generalisation," we are contrasting our method with standard empirical testing. In standard deep RL, test scores are point estimates calculated from samples that offer no formal guarantee for unseen data. PB-SAC, by contrast, computes a rigorous lower bound during training, which certifies generalisation in the sense that this lower bound value guarantees that the policy's expected return on unseen data will not fall below this calculated threshold (Solid line in Panel (b), Figures 1, 4, and 5), proving the agent hasn't simply overfitted to the training data.
> > >
> > > Consequently, when we describe PB-SAC as "safer" than SAC, we are referring to deployment safety measured by the gap between empirical return estimate and lower bound value, rather than other "Safe RL" notions like an agent avoiding physical obstacles. Standard SAC can achieve near-perfect scores in training but fail catastrophically –and silently– upon deployment, offering no warning that it has merely memorised data. PB-SAC is safer because its lower bound acts as a certificate, which can serve as a warning.
> > >
> > > For instance, If a policy yields a high empirical return but a low generalisation certificate value, this is evidence that the policy is not safe to deploy because the high empirical return is not a good measure of future returns on unseen data. Whereas if there is a small gap between a policy's empirical return and its lower bound value, this is evidence that the empirical return estimate is a good measure of future returns on unseen data. By actively giving evidence to assess reliability of the empirical estimate, potentially warning us before a dangerous deployment, PB-SAC replaces blind trust in empirical scores with a quantifiable certificate.
> > >
> > > We thank the reviewer again for the opportunity to clarify this crucial distinction, as it will undoubtedly improve the clarity of our revised manuscript. We hope this better articulates our previous point.

---

### Official Review · Reviewer_iHmT · 2026-03-11

**Soundness:** 3
**Presentation:** 3
**Significance:** 2
**Originality:** 2
**Overall Recommendation:** 4
**Confidence:** 3

**Summary:**

This work introduces a new PAC-Bayesian generalization bound for Reinforcement Learning that aims to quantify the gap between the population risk and the empirical risk. The authors provide a theoretical analysis of their bound, and adapt the SAC algorithm to optimize for the bound, and claim that the algorithm provides competitive performance while guiding exploration and providing confidence certificates. Empirical results on the MuJoCo benchmark are provided to back up the claims.

**Compliance With Llm Reviewing Policy:**

Affirmed.

**Final Justification:**

My main concern about reward propagation in the modified trajectories has been addressed.
Concerns about the empirical results have, for the most part, been addressed. I understand that one advantage of the method is that it provides performance certificates. However, I am not fully convinced by the claimed performance improvements in a sparse-reward setting, as these claims are supported only by experiments in a single environment and do not clearly demonstrate improved performance, especially given the high variance in the return curves.
Overall, I think this work still represents an interesting step toward reinforcement learning algorithms with generalization performance guarantees.

**Key Questions For Authors:**

The key questions are given in the weaknesses for context.

**Limitations:**

yes

**Strengths And Weaknesses:**

Strengths:

This paper addresses the important question of generalization guarantees in RL. Prior PAC bounds proposed were either not suited for the time dependence of RL training data or vacuous. The work builds on prior methods to establish PAC bounds. The main theoretical contribution stems from a bounded difference between empirical losses (Lemma 3.1). Overall the paper is well written and easy to follow.

Weaknesses

- The PAC-Bayesian bound introduced in this study relies on the bounded difference of two empirical losses computed on two datasets that differ only in one tuple (s,a,r,s’) (Lemma 3.1). The proof relies solely on the analysis of how the new reward \bar(r) propagates in the trajectory. However, it seems that changing a tuple in the trajectory should also affect the rest of the trajectory via s’ and \pi(s’), which impacts the likelihood ratio used in the loss computation. This effect is not discussed in the analysis.


- The main weakness of this paper is the empirical analysis, and the gap between empirical results and some of the claims.

 (a) Authors claim PB-SAC improves exploration and guides learning. However, the training curves seem similar to the SAC baseline on all proposed benchmarks and no ablation study is provided to validate this claim. Why does PB-SAC improve exploration? Testing on other environments could help clarify this point.

 (b) Figure 1b displays the gap between the empirical discounted return and the certified discounted return. The graph clearly shows the certificate lower bounds the empirical return, however I don’t understand why the PAC-Bayesian bounds consistently tighten throughout training. The gap between the curves seems constant throughout the training. Maybe the authors could clarify how to interpret these graphs?

(c) Many design choices are not backed up with empirical evidence. In particular, the authors claim that the policy posterior synchronization stabilizes the learning, but no ablation study is performed and the training seems as stable as the other baselines.

- The proposed algorithm, PB-SAC, introduces many computationally intensive steps during the learning to compute the bound and update the policy parameters. The authors should discuss the impact of this extra algorithmic complexity, especially on the training time and required resources.

- Although the main body of the paper is well written, the appendix section needs editing to properly format the figures and pagination. In addition, the main body of the paper mentions results on four benchmark problems, but only two are shown. I would suggest moving the main results of each benchmark to the main body for clarity.

---

> ### Author Rebuttal · Authors · 2026-03-29
>
> # Response to Reviewer iHmT
>
> We sincerely thank the reviewer for their thoughtful review. We are glad they found the paper well-written. We address their specific concerns below.
>
> ## Lemma 3.1 and the Propagation of States (Weakness 1)
>
> The intuition is correct: changing a transition at step $h$ alters $s'$ and (in the worst case) propagates to alter all subsequent states and actions. We explicitly resolve this in App. B.4. We apologize if this was not sufficiently highlighted in the main text (lines 210-216).
>
> For likelihood ratio divergence, we rely on the standard assumption (Sec 3.1, and response to **JQYw**) that weighted rewards remain bounded in $[0, R_{\max}]$.
>
> Because a perturbation at step $h$ affects all subsequent transitions, the bounded-differences indicator $\\mathbb{I}[\\xi^{(j)}\_{h'} \\neq \\bar{\\xi}^{(j)}\_{h'}]$ becomes 1 for every step $h' \\geq h$ in the worst case. In Appendix B.4, we first derive the sum of our coefficients for all these affected subsequent steps. Crucially, if the remainder of the trajectory completely diverges (e.g., the original trajectory receives $0$ weighted reward and the perturbed one receives $R\_{\\max}$ weighted reward at every step), the absolute maximum change in the discounted empirical return exactly matches this sum.
>
> Therefore, our derivation of $\\|c\\|^2$ fully mathematically accounts for the worst-case propagation of errors through the Markov dependency structure. We will move a summary of this proof to the main text to ensure this is clear.
>
> ## Missing Ablations and Exploration Claims (Weaknesses 2a & 2c)
>
> We respectfully point out that several ablation studies and exploration validations are included in the submission, though they were placed in the Appendix due to strict space limits. We completely agree they are critical.
>
> - **Exploration in Sparse Environments (2a):** As shown in Figure 5 (App G.1), we evaluated PB-SAC on the highly challenging sparse-reward task "Ant-v5 (very delayed)" from Tasdighi et al. (2025). The empirical results clearly show that PB-SAC effectively explores and solves the task compared to the baselines.
>
> - **Ablations on Architecture (2c):** Figure 7 (App H) contains the exact ablation studies requested. Figure 7a shows the severe "sawtooth" instability that occurs without our adaptive sampling mechanism, proving its importance for stable learning. Furthermore, Figure 7b relates to Section 4.1 and is largely optional in a dense reward setup; we experimented with different initial $\\epsilon\_{\\text{explore}}$ values, and the performance did not change significantly. We have added this detail for completeness.
>
> - **Policy-Posterior Synchronisation (2c):** The reviewer is correct to highlight this particular point. Synchronisation is an absolute necessity: without it, standard SAC updates would completely detach the actor from the periodically updated posterior mean. This separation would defeat the entire purpose of optimising the PAC-Bayes bound and leveraging its certificate. We will make this explicit in the revision. We acknowledge that this point will benefit from better clarification in the text.
>
>
> ## Clarification on "Tightening" Bounds (Weakness 2b)
>
> The reviewer noted that the absolute gap between the empirical and certified return seems constant. Whilst it may not be immediately visible in Figures 1, 4, and 5(b), the gap consistently tightens throughout the training time steps (Global Steps), tracking the true performance closely. We invite the reviewer to also check Figure 6, where we isolate the mixing time estimation using different _a priori_ fixed values; there, the tightening of the gap throughout the global steps is clearly visible.
>
> ## Extra Algorithmic Complexity and Training Time
>
> We agree that maintaining a posterior is computationally intensive, which is why PB-SAC is designed to aggressively amortize this cost. Heavy bound optimisation and high-rate sampling (256 samples) for critic adaptation occur only once every 20,000 steps (App F, Table 1). Otherwise, PB-SAC runs standard SAC using just the posterior mean.
>
> We will discuss the compute complexity issues, as suggested, while in subsequent work we plan to explore further improvements on posterior design and computational efficiency. That said, we believe the results and their impact potential merit attention of the ICML community, see also our response to **EyBx** on this point.
>
> ## Formatting and Main Text Figures
>
> We apologize for the appendix formatting issues and will correct the pagination and figures. We will also follow the reviewer's suggestion to combine the benchmark plots so all four main experiments appear in the main body (we had originally deferred them to the appendix due to space limits).
>
> We thank the reviewer again for their constructive critique. We hope our response clears up all the issues and that the reviewer might consider raising their score.

---

> > ### Author Rebuttal · Reviewer_iHmT · 2026-04-03
> >
> > I thank the authors for their detailed response. My concerns have been addressed, I will upgrade my score.

---

> > > ### Author Response · Authors · 2026-04-05
> > >
> > > We sincerely thank the reviewer for their time, the constructive feedback, and for upgrading their score. We are glad our response fully addressed their concerns.

---

### Official Review · Reviewer_NZH2 · 2026-03-13

**Soundness:** 3
**Presentation:** 3
**Significance:** 3
**Originality:** 3
**Overall Recommendation:** 4
**Confidence:** 2

**Summary:**

This work derives an improved PAC-Bayesian bound for RL accounting for Markov dependencies via mixing time, achieving an improvment on $(1−γ)^{-1}$ factor on previous results. Then authors also proposed a new algorithm(PB-SAC) for PAC-Bayesian RL.

**Compliance With Llm Reviewing Policy:**

Affirmed.

**Final Justification:**

My major concerns have been addressed by the authors' rebuttal.

**Key Questions For Authors:**

As  mentioned in the weakness part, in practice there are usually clipping operation of the importance weights to ensure stability, does the proposed certificate remain formally valid after such kind of operation?

**Limitations:**

yes

**Strengths And Weaknesses:**

Strengths:
The observation for improving the prevoius PAC-Bayesian bound (i.e., directly bounding the value error through concentration of discounted returns rather than going through the Bellman error) is technically sound and delicate.

Weaknesses:
The treatment of importance weights feels hand-wavy. The paper assumes weighted rewards remain bounded (as authors mentioned in Section 3.1), but clipping importance weights introduces bias, which would invalidate the unbiasedness that the bound relies on.

---

> ### Author Rebuttal · Authors · 2026-03-29
>
> We sincerely thank the reviewer for their positive assessment of our technical contribution.
>
> The reviewer raises an excellent point regarding Importance Sampling (IS) weight clipping. We agree that clipping IS weights introduces bias. However, we would like to clarify that the proposed certificate remains formally valid. Indeed, the bias introduced by clipping is strictly pessimistic. We present the justification below.
>
> By our MDP formulation (Section 2.1), rewards are bounded in $[0, R_{\\max}]$, meaning the trajectory return is strictly non-negative: $G(\\xi) \\ge 0$. (For environments like MuJoCo that can yield negative rewards, we can apply a constant positive shift so all $r_t \\ge 0$; this preserves the pessimistic property below without altering the optimal policy.) Let the clipped importance weight be $w^c_j = \\min(w_j, M)$ for some clipping threshold $M$. Because $G(\\xi) \\ge 0$ and $w^c_j \\le w_j$, the clipped return is strictly less than or equal to the unclipped return: $w^c_j G(\\xi) \\le w_j G(\\xi)$.
>
> Recall that our expected loss is defined as the negative expected return: $\\mathcal{L}(\\theta) = -\\mathbb{E}[w G(\\xi)]$. If we define the true clipped loss as $\\mathcal{L}_c(\\theta) = -\\mathbb{E}[w^c G(\\xi)]$, it follows that:
>
> $$\\mathcal{L}(\\theta) \\le \\mathcal{L}_c(\\theta)$$
>
> When we apply our PAC-Bayesian concentration inequality to the clipped empirical loss $\\hat{\\mathcal{L}}_c(\\theta)$, we obtain a high-probability upper bound on $\\mathcal{L}_c(\\theta)$:
>
> $$\\mathcal{L}_c(\\theta) \\le \\hat{\\mathcal{L}}_c(\\theta) + \\text{Complexity\\_Term}$$
>
> Chaining the inequalities yields:
>
> $$\\mathcal{L}(\\theta) \\le \\mathcal{L}_c(\\theta) \\le \\hat{\\mathcal{L}}_c(\\theta) + \\text{Complexity\\_Term}$$
>
> Therefore, clipping does not invalidate the certificate; it merely makes the lower bound on the expected value slightly more conservative.
>
> _(Note on implementation: we use Softmax Self-Normalised Importance Sampling over standardised log-ratios, which naturally bounds the weights in $[0,1]$ and enjoys similar conservative bounding properties.)_
>
> **Action taken for the revised version:** We agree that the current draft's treatment of this point is insufficiently discussed. In the revised version, we will expand on it in the Appendix to explicitly include the inequalities shown above, which we judge sufficient to rigorously prove that bounded/clipped importance weights maintain the formal validity of the generalisation certificate.
>
> We thank the reviewer again for the feedback, helping us strengthen the theoretical grounding of the paper. We hope this clarification addresses their concern and that they might consider raising their score.

---

> > ### Author Rebuttal · Reviewer_NZH2 · 2026-04-03
> >
> > Thank you for the rebuttal. My concerns have been addressed.

---

> > > ### Author Response · Authors · 2026-04-05
> > >
> > > We sincerely thank the reviewer for their time and for confirming that all of their concerns have been fully addressed. We are very grateful for their constructive feedback and the effort they invested in helping us improve the paper.

---

### Decision · Program_Chairs · 2026-04-30

**Decision:**

Accept (regular)

**Comment:**

The paper derives a new improved PAC-Bayesian bound for RL, propose a new Bayesian algorithm for deep RL and evaluates it on continuous control tasks.

The reviewers agree that the theoretical contribution is significant and that the empirical results are promising.
Several important issues raised by the reviewers were fully solved during the discussion phase.
For this reason, I urge the authors to incorporate the following improvements in the final version, using the extra page as needed:

- Weight clipping and pessimistic bias: this is very important, it should be made at least as clear in the paper as it was in your rebuttal
- Reward propagation in the modified trajectories: this also should be made more clear in the main text
- Motivation for the algorithm and a clear explanation of your concept of safety
- The meaning of "non independent trajectories" should be made unambiguous
- Tractability of PGE should be fully discussed in the paper